# Improving Diffusion Planners by Self-Supervised Action Gating with Energies

Yuan Lu [1]   Dongqi Han [2]   Yansen Wang [2]   Dongsheng Li [2]

## Abstract

Diffusion planners are a strong approach for offline reinforcement learning, but they can fail when value-guided selection favours trajectories that score well yet are locally inconsistent with the environment dynamics, resulting in brittle execution. We propose Self-supervised Action Gating with Energies (SAGE), an inference-time re-ranking method that penalises dynamically inconsistent plans using a latent consistency signal. SAGE trains a Joint-Embedding Predictive Architecture (JEPA) encoder on offline state sequences and an action-conditioned latent predictor for short horizon transitions. At test time, SAGE assigns each sampled candidate an energy given by its latent prediction error and combines this feasibility score with value estimates to select actions. SAGE can integrate into existing diffusion planning pipelines that can sample trajectories and select actions via value scoring; it requires no environment rollouts and no policy re-training. Across locomotion, navigation, and manipulation benchmarks, SAGE improves the performance and robustness of diffusion planners.

## 1. Introduction

Reinforcement learning (RL) from a fixed, offline dataset has become a central paradigm for sequential decision making in robotics and artificial intelligence (Bellman, 1957; Sutton et al., 1998; Silver et al., 2017). By reusing demonstrations, offline RL promises to acquire complex behaviours without the cost, delay, or safety risks of online trial-and-error (Levine et al., 2020; Fujimoto et al., 2019). Yet the absence of online interaction also makes offline RL fragile: once a learnt policy deviates from the behaviour distribution, estimation errors can compound, and performance can deteriorate sharply in long-horizon tasks.

A recent line of work reframes offline control as conditional generative modelling of state-action trajectories. Instead of outputting a single action, these methods model a distribution over action sequences conditioned on the current state, then plan by sampling (Ha & Schmidhuber, 2018). Diffusion models are particularly appealing in this role because they can represent complex, multimodal sequence distributions and generate diverse candidates through iterative denoising (Ho et al., 2020; Sohl-Dickstein et al., 2015). Diffusion-based trajectory planners such as Diffuser (Janner et al., 2022) operationalise this idea by learning a trajectory diffusion model and using guidance or scoring to bias sampling toward high-return futures, often in a receding-horizon loop (Chi et al., 2025; Ajay et al., 2022).

Despite their empirical success, diffusion planners exhibit a failure mode that is easy to miss in value-centric formulations. Planning typically proceeds by sampling many candidate futures, scoring them with a learnt value model, and selecting the best-looking candidate. This approach can inadvertently reward trajectories that are attractive under the scorer yet are locally inconsistent with feasible dynamics (Ki et al., 2025; Dong et al., 2024a). When a plan's early transitions cannot be realised from the current state by any action sequence, execution becomes fragile: the agent commits to an unrealistic prefix, and the resulting mismatch can cascade under replanning (Zhou et al., 2023).

In this work, we argue that feasibility should be treated as a signal distinct from value in diffusion-based decision making, rather than being implicitly absorbed by a single critic. A natural response is to bake feasibility into the generative process via auxiliary guidance, constraints, or verifiers (Lee et al., 2023; Wang et al., 2022; Ding & Jin, 2023; Ada et al., 2024; Zhang et al., 2024), but these often require additional models and objectives, which can increase training complexity and limit scalability.

Our perspective is inspired by recent large-scale empirical evidence showing that, for diffusion planning, unguided sampling followed by candidate ranking can outperform guided sampling (Lu et al., 2025; Wang et al., 2025; Chen & Fleming, 2025; Zhu et al., 2023; Hansen-Estruch et al., 2023; Ki et al., 2025; Chi et al., 2025). Recent studies suggest that injecting guidance during denoising may distort the

Work done during an internship of Yuan Lu at Microsoft Research [1]University College London, London, UK [2]Microsoft Research. Correspondence to: Dongqi <dongqi-han@microsoft.com>.

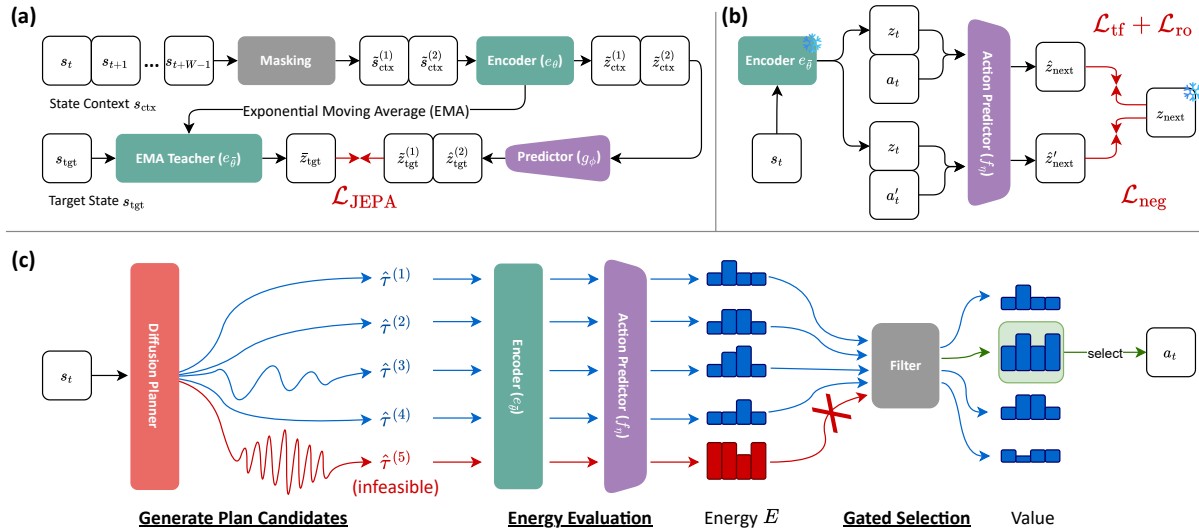

*Figure 1.* The SAGE framework. **(a)** Learn a predictive JEPA state representation from masked state windows with an EMA teacher. **(b)** Train an action-conditioned latent predictor whose prediction error defines a transition energy. **(c)** At test time, score diffusion-generated candidate plans with this energy and gate value-based selection toward locally feasible actions

learnt trajectory distribution, placing increased importance on the selection stage (Mao et al., 2024; Park et al., 2025). However, ranking-based planners still rely on a single critic that is implicitly tasked with two competing roles: identifying high-value futures and rejecting locally infeasible trajectories. These objectives can be in tension in offline settings, as value optimisation encourages extrapolation beyond the dataset, while feasibility requires conservatism and adherence to in-distribution dynamics.

**Contributions.** Motivated by this tension, we propose Self-supervised Action Gating with Energies (SAGE), a selector-side factorisation of decision signals for diffusion planning. SAGE explicitly separates feasibility from value by introducing a self-supervised feasibility score that evaluates the local feasibility of candidate trajectory prefixes, while leaving value estimation unchanged. Unlike prior verifier or guidance approaches, SAGE learns feasibility via predictive self-supervision without requiring negative sampling strategies or environment interaction, making it naturally scalable to large and diverse offline datasets (Assran et al., 2023; Liu et al., 2021; Laskin et al., 2020).

SAGE consists of two learnt components trained purely offline: (i) a Joint-Embedding Predictive Architecture (JEPA) encoder trained on state sequences to capture dataset-consistent dynamics, and (ii) an action-conditioned latent predictor that models short-horizon transitions in the learnt latent space. At inference time, SAGE scores each candidate trajectory using a latent consistency energy and combines this score with the planner's original value-based selection. This design can work with guided or unguided sampling and requires no retraining of the diffusion planners.

## 2. Background

### 2.1. Offline Reinforcement Learning

We consider reinforcement learning from a fixed dataset collected by an unknown behaviour policy. The environment is modelled as a discounted Markov decision process (MDP) with state space $\mathcal{S}$, action space $\mathcal{A}$, transition kernel $\mathcal{T}(s' \mid s, a)$, reward function $r(s, a, s')$, and discount factor $\gamma \in [0, 1)$ (Sutton et al., 1998). In offline RL, the agent is given a dataset $\mathcal{D} = \{(s_t, a_t, r_t, s_{t+1})\}_{t=1}^{N}$ and must learn to act without additional interaction with the environment (Fujimoto et al., 2019; Levine et al., 2020).

A central challenge in offline RL is *distributional mismatch*. When a learnt policy proposes actions or trajectories that are weakly supported by the dataset, errors in value estimation and dynamics can compound, leading to brittle behaviour (Kumar et al., 2019; 2020). This issue persists even when powerful function approximators are used, motivating alternative formulations of offline control.

### 2.2. Diffusion-based Decision Making

We can cast offline control as a conditional generative modelling problem: rather than learning a pointwise policy $\pi(a \mid s)$, we learn a model that generates action sequences or trajectories conditioned on the current state (Ha & Schmidhuber, 2018). Diffusion models are effective here because they represent complex multimodal sequence distributions and sample diverse candidates via iterative denoising (Sohl-Dickstein et al., 2015; Ho et al., 2020).

A convenient unifying view is to generate a length-$H$ se-

quence $\tau$ conditioned on $s_t$. Two common instantiations are: **(i) Diffusion planners** model trajectory distributions:

$$\tau = \begin{bmatrix} s_t & s_{t+1} & \cdots & s_{t+H} \\ a_t & a_{t+1} & \cdots & a_{t+H-1} \end{bmatrix} \quad \text{or} \quad [s_t, s_{t+1}, \ldots, s_{t+H}], \quad (1)$$

and act by sampling candidate futures and selecting among them in a receding-horizon loop (Janner et al., 2022; Ajay et al., 2022; Chi et al., 2025). **(ii) Diffusion policies** model actions directly,

$$p_\theta(a_t \mid s_t) \quad \text{or} \quad p_\theta(a_{t:t+H-1} \mid s_t), \quad (2)$$

often trained end-to-end with offline RL objectives and regularisation (Wang et al., 2022; Kang et al., 2023)

### 2.3. Candidate Generation and the Feasibility Gap

A diffusion planner can be abstracted as a candidate generator. Given conditioning information $c_t$ (e.g., current state $s_t$), it samples a set of $C$ candidate trajectories

$$\hat{\tau}_t^{(1:C)} \sim p_\theta(\tau \mid s_t), \quad (3)$$

where each candidate $\hat{\tau}_t^{(i)} = (\hat{s}_{t:t+H}^{(i)}, \hat{a}_{t:t+H-1}^{(i)})$ is a horizon-$H$ plan. Control is performed by scoring candidates with a learnt value or return surrogate $J_\psi$ and executing the first action of the best-scoring candidate:

$$\hat{\tau}_t^* \in \arg\max_{i \in [C]} J_\psi(\hat{\tau}_t^{(i)}), \qquad a_t \leftarrow \hat{a}_t^{(*)}. \quad (4)$$

While effective, this "generate-many, pick-one" procedure introduces a subtle failure mode. The scoring function $J_\psi$ evaluates long-term desirability, but does not explicitly assess whether the candidate trajectory is locally executable from the current state. As a result, candidates that score highly may contain locally inconsistent or dynamically implausible first steps, leading to brittle execution under replanning (Dong et al., 2024a; Zhou et al., 2023). This motivates treating feasibility as a signal distinct from value.

### 2.4. Self-supervised Predictive Representations

Self-supervised learning provides a natural mechanism for modelling which transitions are typical under dataset dynamics, without relying on rewards or environment rollouts. Joint-Embedding Predictive Architectures (JEPAs) learn representations by predicting the latent embedding of a future or masked input from a context window, operating entirely in representation space rather than reconstructing observations (LeCun, 2022; Assran et al., 2023).

Let $e_\theta$ denote an encoder, $e_{\bar{\theta}}$ an exponential-moving-average (EMA) teacher, and $g_\phi$ a predictor. Given a context input $s_{\text{ctx}}$ and a target input $s_{\text{tgt}}$, a typical JEPA objective is

$$\mathcal{L}_{\text{JEPA}} = \left\| g_\phi(e_\theta(s_{\text{ctx}})) - \text{sg}(e_{\bar{\theta}}(s_{\text{tgt}})) \right\|_2^2. \quad (5)$$

When trained on sequential data, low prediction error indicates that the target transition is predictable from context under the dataset dynamics, while high error signals an atypical or out-of-support transition (Schwarzer et al., 2020).

### 2.5. Local Transition Consistency as a Feasibility Signal

For planning, we are not concerned with global trajectory likelihoods, but with whether the early prefix of a candidate plan is locally executable from the current state (Chua et al., 2018). To capture this notion, we consider short-horizon predictive consistency in latent space. Let $z_t = e(s_t)$ be the latent encoding of a state, and let $f_\eta$ be an action-conditioned latent predictor trained on offline transitions:

$$\hat{z}_{t+1} = f_\eta(z_t, a_t). \quad (6)$$

Discrepancies between predicted latents and planned latents over a short prefix provide a scalar measure of local transition inconsistency. Aggregating this discrepancy yields an energy-like score that reflects how well a candidate aligns with dataset-supported dynamics (Du & Mordatch, 2019). In the next section, we introduce how to use this signal as an inference-time gating mechanism for diffusion planners.

## 3. Self-Supervised Action Gating

**Self-supervised Action Gating with Energies (SAGE)** is designed to be a inference-time mechanism that augments diffusion-based planners with an explicit notion of local feasibility. SAGE is trained entirely from offline trajectories and does not modify or retrain the diffusion planner. Instead, it learns a self-supervised feasibility signal that reranks sampled plans at inference time, as depicted in Figure 1. SAGE decomposes planning into two components: (i) a *candidate generator* and (ii) an *energy-based selector*, which penalises plans whose early prefixes are locally inconsistent with dataset dynamics. The feasibility signal is implemented as a latent consistency energy learnt via a two-stage training pipeline: First, we learn a predictive latent state representation using self-supervision. Second, we learn an action-conditioned predictor that models short-horizon transitions in this latent space. Both stages use only offline data and require no reward or environment interaction.

### 3.1. Training Predictive State Representation

The goal of Stage I is to learn a latent space in which dataset-consistent future states are predictable from local context. We train a JEPA Encoder on state-only trajectories. Let $e_\theta$ be a state encoder and $e_{\bar{\theta}}$ an exponential-moving-average (EMA) teacher. From an offline trajectory, we sample a context window $s_{\text{ctx}} = (s_t, \ldots, s_{t+W-1})$ and a set of future offsets $\mathcal{K} \subseteq \{1, \ldots, \mathcal{K}_{\max}\}$. For each $k \in \mathcal{K}$, the corresponding future target is defined as $s_{\text{tgt}}^{(k)} \triangleq s_{t+W-1+k}$.

**Algorithm 1** Encoder Pre-Training

---

**Require:** Offline data $\mathcal{D}$, window $W$, offsets $\mathcal{K}$
 1: Initialise Encoder $e_\theta$, EMA teacher $e_{\bar\theta}$, predictor $g_\phi$
 2: **for** each step **do**
 3:     Sample $t$ and $\mathcal{K}$; set $s_{\mathrm{ctx}} \leftarrow (s_t, \ldots, s_{t+W-1})$
 4:     Views: $\tilde{s}_{\mathrm{ctx}}^{(1)}, \tilde{s}_{\mathrm{ctx}}^{(2)} \leftarrow \mathrm{Mask}(s_{\mathrm{ctx}})$
 5:     Targets: $s_{\mathrm{tgt}}^{(k)} \leftarrow s_{t+W-1+k}$ for $k \in \mathcal{K}$
 6:     $z_{\mathrm{ctx}}^{(i)} \leftarrow e_\theta(\tilde{s}_{\mathrm{ctx}}^{(i)}), \quad \bar{z}_{\mathrm{tgt}}^{(k)} \leftarrow e_{\bar\theta}(s_{\mathrm{tgt}}^{(k)})$
 7:     $\hat{z}_{\mathrm{tgt}}^{(i,k)} \leftarrow g_\phi(z_{\mathrm{ctx}}^{(i)}, k)$
 8:     Incur loss $\mathcal{L} \leftarrow \sum_{i\in\{1,2\}, k\in\mathcal{K}} \mathcal{L}_{\mathrm{JEPA}}(\hat{z}_{\mathrm{tgt}}^{(i,k)}, \bar{z}_{\mathrm{tgt}}^{(k)})$
 9:     Update $\theta, \phi$; update EMA $\bar\theta$
10: **end for**
11: **Return** frozen $e_{\bar\theta}$

---

We construct two stochastic context views $\tilde{s}_{\mathrm{ctx}}^{(1)}, \tilde{s}_{\mathrm{ctx}}^{(2)}$ by randomly masking state features and time-steps, while keeping targets uncorrupted. A predictor $g_\phi$ is trained to map the latent context to the teacher embedding of the future targets using a mask-token Transformer readout (see Appendix C):

$$z_{\mathrm{ctx}}^{(i)} = e_\theta(\tilde{s}_{\mathrm{ctx}}^{(i)}), \quad \bar{z}_{\mathrm{tgt}}^{(k)} = e_{\bar\theta}(s_{\mathrm{tgt}}^{(k)}), \quad \hat{z}_{\mathrm{tgt}}^{(i,k)} = g_\phi(z_{\mathrm{ctx}}^{(i)}, k).$$

We minimise a self-supervised prediction objective that aligns $\hat{z}_{\mathrm{tgt}}^{(i,k)}$ with $\bar{z}_{\mathrm{tgt}}^{(k)}$ in latent space, together with VICReg regularisation to prevent collapse (Bardes et al., 2021). After training, we freeze the EMA encoder $e_{\bar\theta}$ and use it as the representation function for action-conditioned predictor training. Algorithm 1 summarises the Encoder training procedure. Architectural and VICReg regularisation details can be found in Appendix C.1.

### 3.2. Action-Conditioned Latent Predictor

In Stage II, we learn an action-conditioned short-horizon predictor in the frozen JEPA latent space. We fix the EMA encoder $e_{\bar\theta}$ and embed offline windows $(s_{t:t+W}, a_{t:t+W-1})$ into latents $z_{t:t+W} = e_{\bar\theta}(s_{t:t+W})$. The predictor $f_\eta$ is implemented as a block-causal Transformer over latent–action tokens and outputs one-step predictions $\hat{z}_{t+1:t+W}$. We train $f_\eta$ with three complementary objectives.

(i) **Teacher-forced one-step loss** encourages accurate next-latent prediction under ground-truth prefixes:

$$\mathcal{L}_{\mathrm{tf}} = \sum_{j=0}^{W-1} \left\| \hat{z}_{t+1+j} - z_{t+1+j} \right\|_1. \tag{7}$$

(ii) **Short-horizon rollout loss** enforce consistency under autoregressive application of $f_\eta$ for a small horizon $H_{\mathrm{ro}}$:

$$\mathcal{L}_{\mathrm{ro}} = \left\| \hat{z}_{t+H_{\mathrm{ro}}} - z_{t+H_{\mathrm{ro}}} \right\|_1, \tag{8}$$

where $\hat{z}_{t+H_{\mathrm{ro}}}$ is obtained by rolling out $f_\eta$ from $z_t$ using the action sequence $a_{t:t+H_{\mathrm{ro}}-1}$.

(iii) **Action-usage hinge** discourages predictions that remain accurate when actions are mismatched. We permute actions within the batch to form $a'$ (see Appendix C.2), compute the corresponding prediction error $E_{\mathrm{neg}} = \sum_j \|\hat{z}'_{t+1+j} - z_{t+1+j}\|_1$, and apply a margin:

$$\mathcal{L}_{\mathrm{neg}} = \left[ m - E_{\mathrm{neg}} \right]_+. \tag{9}$$

The final objective combines the three terms:

$$\mathcal{L}_{\mathrm{AC}} = \mathcal{L}_{\mathrm{tf}} + \lambda_{\mathrm{ro}}\mathcal{L}_{\mathrm{ro}} + \lambda_{\mathrm{neg}}\mathcal{L}_{\mathrm{neg}}. \tag{10}$$

After training, $f_\eta$ is frozen and used to compute the consistency energy in Section 3.3 (Eq. 11).

### 3.3. Inference

SAGE is used only at inference time. At each decision step $t$, a diffusion planner samples $C$ candidate plans $\{\hat{\tau}_t^{(i)}\}_{i=1}^C \sim p_\theta(\tau \mid s_t)$, where each $\hat{\tau}_t^{(i)} = (\hat{s}_{t:t+H}^{(i)}, \hat{a}_{t:t+H-1}^{(i)})$ is a horizon-$H$ trajectory, If the base planner outputs states only, we infer the required short action prefix via an inverse dynamics model (Agrawal et al., 2016). SAGE evaluates each candidate using a short prefix of length $K \ll H$ and prefers candidates whose early transitions are locally predictable under dataset dynamics. Let $z = e_{\bar\theta}(s)$ denote the frozen latent encoding from Stage I, and let $f_\eta$ be the action-conditioned predictor from Stage II. For candidate $i$, we define the latent consistency energy over the first $K$ transitions as:

$$E(\hat{\tau}_t^{(i)}) = \frac{1}{K} \sum_{k=0}^{K-1} \left\| f_\eta\big(z_{t+k}^{(i)}, a_{t+k}^{(i)}\big) - z_{t+k+1}^{(i)} \right\|_1. \tag{11}$$

Low energy indicates that the candidate prefix follows dataset-consistent local dynamics. We select candidates by combining energy with an existing planner score $J$. We keep the lowest-energy $\mathcal{P}$ fraction of candidates and then choose the best remaining candidate via a soft penalty:

$$i^* \in \arg\max_{i\in\mathcal{I}_t} \left( J(\hat{\tau}_t^{(i)}) - \lambda E(\hat{\tau}_t^{(i)}) \right), \quad a_t \leftarrow \hat{a}_t^{(i^*)}, \tag{12}$$

where $\mathcal{I}_t$ is the set retained after energy filtering. SAGE adds $\mathcal{O}(CK)$ lightweight encoder/predictor evaluations per decision step. Algorithm 2 summarises the resulting inference loop. We report wall-clock overhead in Appendix D.2

## 4. Experiments

SAGE is designed as an inference-time *feasibility selector*: it penalises sampled candidates whose early prefixes are locally inconsistent with dataset dynamics. Our experiments address two questions: (i) does the latent consistency energy behave like an *action-conditioned feasibility signal*? (ii) does SAGE improve end-to-end control across

**Algorithm 2** SAGE Inference

**Require:** Planner $p_\theta$, encoder $e_{\bar{\theta}}$, predictor $f_\eta$, score $J$, prefix $K$, keep-rate $\mathcal{P}$, penalty $\lambda$
1: **for** each decision step $t$ **do**
2:     Sample $\{\hat{\tau}_t^{(i)}\}_{i=1}^C \sim p_\theta(\tau \mid s_t)$
3:     Compute $\{E(\hat{\tau}_t^{(i)})\}_{i=1}^C$ on the first $K$ steps (Eq. 11)
4:     Keep $\mathcal{I}_t \leftarrow$ lowest-energy $\mathcal{P}$ fraction
5:     Choose $i^* \leftarrow \arg\max_{i \in \mathcal{I}_t} \left( J(\hat{\tau}_t^{(i)}) - \lambda E(\hat{\tau}_t^{(i)}) \right)$
6:     Execute $a_t \leftarrow \hat{a}_t^{(i^*)}$ and observe $s_{t+1}$
7: **end for**

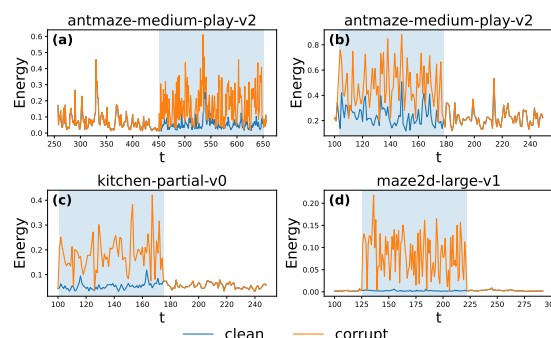

*Figure 2.* SAGE energy localises feasibility violations. Per-step latent-consistency energy on an offline episode with clean and after corrupted action segment; corruption induces a sharp, local spike (shaded interval). Full D4RL results in Appendix D.3 Figure 10.

diverse domains? We first validate the energy with controlled diagnostics, then report benchmark results on D4RL and ablations.

### 4.1. Experiment Setup

**Tasks.** We evaluate on a diverse set of D4RL (Fu et al., 2020) environments spanning locomotion (MuJoCo), navigation (AntMaze, Maze2D), and manipulation (Kitchen). The complete environment list and any task-specific details are provided in Appendix B.

**Baselines.** We compare to representative offline-RL and diffusion decision-making methods: (i) imitation learning: BC; (ii) non-diffusion offline RL: BCQ (Fujimoto et al., 2019), CQL (Kumar et al., 2020), IQL (Kostrikov et al., 2021); (iii) diffusion policies: DQL (Wang et al., 2022) and IDQL (Hansen-Estruch et al., 2023); (iv) diffusion planners: Diffuser (Janner et al., 2022) and the state-of-the-art generate-and-rank planner DV (Lu et al., 2025).

To contextualise SAGE against prior feasibility oriented diffusion planners, we additionally include: RGG (Lee et al., 2023), which learns a restoration-gap predictor and uses it as denoising-time guidance to detect and refine infeasible plans; LDCQ (Venkatraman et al., 2023), which performs batch-constrained planning by sampling latent trajectory candidates from a diffusion prior and selecting them with a learned Q-function; and LoMAP (Lee & Choi, 2025), a training-free test-time projection that pulls intermediate diffusion samples back onto a locally approximated data manifold. These methods represent a comprehensive set of strong current strategies for reliability.

In our experiments, we adopt DV's design for both the SAGE candidate generator and utility scorer $J$. We choose DV's design because it's Monte-Carlo Sampling with Selection (MCSS) generates many unguided trajectories and then selects using a critic, this is exactly the regime where SAGE should help most by improving selection feasibility without changing generation. Using DV's design makes direct comparisons possible. We reproduce baselines marked with * using CleanDiffuser (Dong et al., 2024b) to ensure

consistent architectures and evaluation, other baselines are reported from literature. Code available at Section A

**Evaluation protocol.** All reported numbers use 500 evaluation episode seeds per task. We report D4RL normalised scores where applicable. (see Appendix B).

### 4.2. Relationship between Feasibility and Energy

We first test whether latent consistency energy acts as a local feasibility signal by taking dataset trajectories and corrupting a short action window while leaving the surrounding trajectory unchanged. Given a segment $\{(s_t, a_t)\}_{t=0}^{H-1}$, we sample a window $[t_0, t_1]$ and shuffle actions within it, preserving marginal action realism but breaking temporal alignment with the state sequence to induce a targeted local dynamics inconsistency.

We then compute the per-step energy $e_t$ from the action-conditioned predictor and compare energies inside vs. outside the interval. As shown in Figure 2 across tasks, corruption produces a clear, localised energy spike that aligns with the shuffled window, while energies outside agrees to the clean trajectory. Quantitatively, treating $e_t$ as an anomaly score yields strong localisation AUROC: 0.98 (MuJoCo), 0.98 (Kitchen), 0.99 (Maze2D), and 0.94 (AntMaze), consistent with a feasibility signal sensitive to action-conditioned transition consistency. Additional details and experiments are in Appendix D.1 and Appendix D.3 (Figure 10).

To evaluate the effectiveness of SAGE beyond synthetic corruption, we run a simple stress-test in Maze2D with an intentionally large candidate pool ($C = 500$). So MCSS is more likely to select spuriously high-scoring but impossible plans, as shown in Figure 3 (left). SAGE leaves the generator and critic unchanged, but adds prefix energy filtering and a soft energy penalty at selection time; this suppresses infeasible winners while retaining diverse, corridor-respecting trajectories (right).

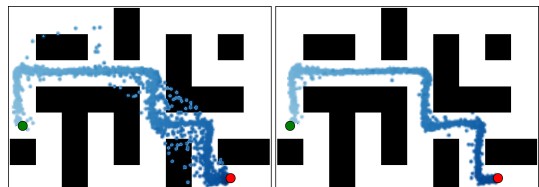

*Figure 3.* 100 trajectories example sampled from Maze2d, MCSS can selects wall-crossing or out-of-bounds trajectories (left), while SAGE's prefix energy filtering and soft penalty suppresses these failures without collapsing trajectory diversity (right).

*Table 1.* **MuJoCo locomotion results for Diffusion Planners.** Values are means over 500 evaluation seeds; standard errors are omitted here for readability, we report the detailed $\pm$ SE in Appendix D.5 Table 13. * denotes results reproduced by us, others are reported from the literature,

| Dataset | Env | Diffuser* | RGG | LoMAP | LDCQ | DV* | SAGE (Ours) |
|---------|-----|-----------|-----|-------|------|-----|-------------|
| M-E | HC | 88.9 | 90.8 | 91.1 | 90.2 | 92.7 | **95.4** |
| M-E | Hop | 103.3 | 109.6 | 110.6 | **113.3** | 108.8 | **111.3** |
| M-E | W2d | 106.9 | 107.8 | 109.2 | 109.3 | 108.6 | **109.4** |
| M | HC | 42.8 | 44.0 | 45.4 | 42.8 | 50.4 | **51.6** |
| M | Hop | 74.3 | 82.5 | **93.7** | 66.2 | 80.9 | 83.9 |
| M | W2d | 79.6 | 81.7 | 79.9 | 69.4 | 82.8 | **84.8** |
| M-R | HC | 37.7 | 41.0 | 39.1 | 41.8 | 45.8 | **46.5** |
| M-R | Hop | 93.6 | 95.2 | **97.6** | 86.3 | 91.6 | 91.8 |
| M-R | W2d | 70.6 | 78.3 | 78.7 | 68.5 | 84.1 | **85.3** |
| **Average** | | 77.5 | 81.2 | 82.8 | 81.6 | 82.9 | **84.4** |

### 4.3. Locomotion in Diffusion Planners

Table 1 reports MuJoCo locomotion results for diffusion planners. Overall, feasibility-oriented baselines outperform Diffuser, suggesting that even in dense-reward locomotion, performance is still limited by brittle sampled candidates that lead to execution failures. Among these, DV* is the strongest baseline, highlighting the need for newer approaches that work in synergy with strong MCSS-style planners. SAGE improves the locomotion average from 82.9 (DV*) to 84.4 (+1.5) without retraining the diffusion generator or critic. SAGE also matches or exceeds earlier planners and prior feasibility variants. Together with the controlled energy diagnostic (Figure 2), these gains are consistent with filtering locally inconsistent early prefixes while preserving DV*'s long-horizon value-based scoring.

### 4.4. Manipulation and Navigation Improvements

Table 2 reports performance across manipulation (Kitchen), navigation with sparse rewards (AntMaze), and long-horizon navigation (Maze2D). These domains expose feasibility failures more sharply than locomotion: plans can become invalid due to compounding model errors, wall-crossings, or locally inconsistent transitions that derail goal-reaching. SAGE achieves the highest performace in all three categories. Across domain, SAGE matches or improves upon the strongest baseline DV*.

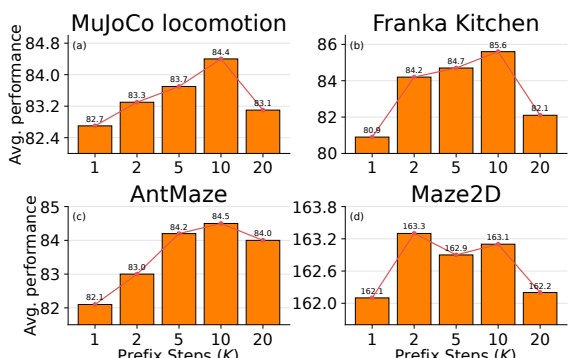

*Figure 4.* Average performance as a function of the prefix window length $K$ across four evaluation domains.

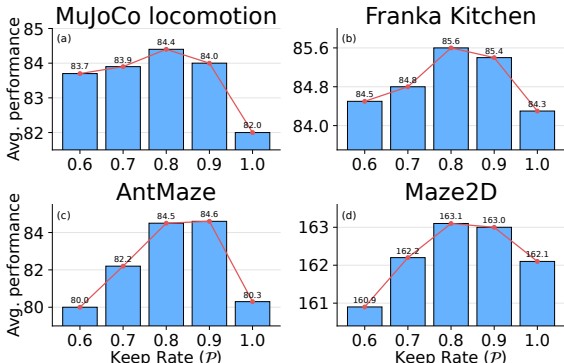

*Figure 5.* Average performance as a function of the keep rate $\mathcal{P}$ across four evaluation domains.

Across navigation tasks, all feasibility-focused planners improve substantially over Diffuser*, particularly on Maze2D where artifact trajectories are common. Nevertheless, strong generate-and-rank planners remain critical for high returns: DV* is consistently stronger than feasibility correction alone, indicating that global task success is not resolved by feasibility filtering by itself.

SAGE improves upon DV* across all three families. In Kitchen, SAGE increases DV* on both datasets (Mixed: $73.6 \rightarrow 74.5$, Partial: $90.0 \rightarrow 96.6$), outperforming LDCQ and diffusion-policy baselines and suggesting that locally inconsistent prefixes remain a failure mode even when the task is dominated by multi-step manipulation. In AntMaze, SAGE improves DV* on all four tasks (category average $81.6 \rightarrow 84.5$), indicating that feasibility-aware selection complements value-based ranking under sparse rewards; LoMAP and LDCQ provide partial gains over Diffuser* but remain far below DV*, consistent with the need for strong long-horizon selection in addition to feasibility control. In Maze2D, where good methods already achieve near optimal scores, SAGE yields smaller but consistent gains overall ($161.6 \rightarrow 163.1$), with the clearest headroom on the Large maze where long-horizon planning is most brittle.

*Table 2.* **Normalised performance of various offline-RL methods on the D4RL benchmark.** Reported values are means over 500 episode seeds; * denotes results reproduced by us; remaining numbers are reported from the literature, — indicates the task was not reported in the original work. Standard errors can be found in the detailed table provided in Appendix D.5 Table 13.

| Dataset | Env | IL | Non-Diffusion | | | Diffusion Policies | | Diffusion Planners | | | | | |
|---|---|---|---|---|---|---|---|---|---|---|---|---|---|
| | | BC | BCQ | CQL | IQL | DQL* | IDQL* | Diffuser* | RGG | LoMAP | LDCQ | DV* | SAGE (Ours) |
| M | Kitchen | 47.5 | 8.1 | 51.0 | 51.0 | 55.1 | 66.5 | 52.5 | — | — | 62.3 | 73.6 | **74.5** |
| P | Kitchen | 33.8 | 18.9 | 49.8 | 46.3 | 65.5 | 66.7 | 55.7 | — | — | 67.8 | 90.0 | **96.6** |
| **Average** | | 40.7 | 13.5 | 50.4 | 48.7 | 60.3 | 66.6 | 54.1 | — | — | 65.1 | 81.8 | **85.6** |
| Antmaze-L | D | 0.0 | 2.2 | 61.2 | 47.5 | 70.6 | 67.9 | 27.3 | — | 39.3 | 57.7 | 76.0 | **77.0** |
| Antmaze-L | P | 0.0 | 6.7 | 53.7 | 39.6 | 81.3 | 63.5 | 17.3 | — | 20.7 | — | 76.4 | **82.1** |
| Antmaze-M | D | 0.0 | 0.0 | 15.8 | 70.0 | 82.6 | 84.8 | 2.0 | — | 36.0 | 68.9 | 85.1 | **88.0** |
| Antmaze-M | P | 0.0 | 0.0 | 14.9 | 71.2 | 87.3 | 84.5 | 6.7 | — | 40.7 | — | 89.0 | **91.0** |
| **Average** | | 0.0 | 2.2 | 36.4 | 57.1 | 80.5 | 75.2 | 13.3 | — | 34.2 | — | 81.6 | **84.5** |
| Maze2D | L | 5.0 | 6.2 | 12.5 | 58.6 | 186.8 | 90.1 | 123.0 | 148.3 | 151.9 | 150.1 | 197.4 | **200.6** |
| Maze2D | M | 30.3 | 8.3 | 5.0 | 34.9 | **152.0** | 89.5 | 121.5 | 130.0 | 131.0 | 125.3 | 150.7 | 150.8 |
| Maze2D | U | 3.8 | 12.8 | 5.7 | 47.4 | **140.6** | 57.9 | 113.9 | 128.3 | 126.0 | 134.2 | 136.8 | 137.9 |
| **Average** | | 13.0 | 9.1 | 7.7 | 47.0 | 159.8 | 79.2 | 119.5 | 135.5 | 136.3 | 136.5 | 161.6 | **163.1** |

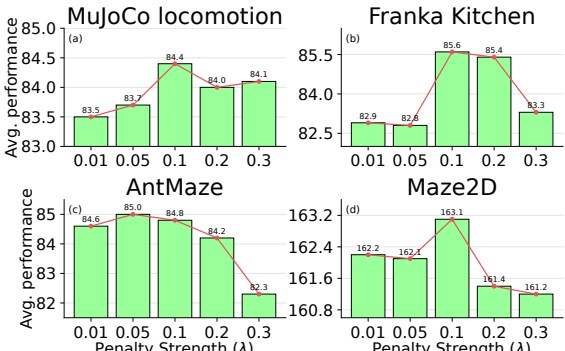

*Figure 6.* Average performance as a function of the penalty strength $\lambda$ across four evaluation domains.

*Table 3.* **Statistical significance tests of SAGE vs. DV.** We test the task-average improvement $\Delta$ within each domain using result of 500 evaluation episode seeds per task. We report the conservative unpaired ($\rho=0$) sensitivity.

| Domain | # | DV* | SAGE | $\Delta$ (95% CI) | $p$ |
|---|---|---|---|---|---|
| Overall | 18 | 95.59 | 97.69 | 2.10 [1.42,2.78] | $1.1\times10^{-9}$ |
| MuJoCo | 9 | 82.86 | 84.44 | 1.59 [1.16,2.02] | $4.8\times10^{-13}$ |
| Kitchen | 2 | 81.80 | 85.55 | 3.75 [3.21,4.29] | $1.1\times10^{-42}$ |
| AntMaze | 4 | 81.62 | 84.53 | 2.90 [0.46,5.34] | 0.020 |
| Maze2D | 3 | 161.63 | 163.10 | 1.47 [-0.54,3.47] | 0.152 |

Overall, these results highlight a consistent picture: prior feasibility-oriented diffusion planners reduce catastrophic artifacts by intervening during sampling or by constraining the planning manifold, while SAGE targets a complementary failure mode: local feasibility of the early prefix and therefore steadily improves over a state-of-the-art planner while preserving its strong generator and critic. We evaluate the significance of SAGE vs. DV* using an unpaired two-sample test on 500 per-episode returns per task (Table 3), see Appendix Section D.4 for details. SAGE shows consistent and statistically significant improvements across D4RL, with the only exception being Maze2D, where gains are attenuated by ceiling effect on Medium and Umaze.

### 4.5. Ablation Studies

SAGE injects a local feasibility signal into value-based selection via three inference-time hyperparameters: the prefix length $K$, the keep-rate $\mathcal{P}$, and the penalty weight $\lambda$. Figures 4–6 illustrate their qualitative effects. In Figure 4,

performance typically improves from very short to moderate prefix windows, while overly long windows (e.g., $K \geq 20$) can degrade performance as prediction errors compound and the energy begins to penalise benign downstream mismatches or averaging away brief infeasible segments in prefix. Figure 5 shows that filtering too aggressively can be over-conservative by reducing candidate diversity and discarding high-value plans, whereas keeping all candidates weakens the feasibility signal and approaches DV's value-only ranking. Finally, Figure 6 shows the trade-off set by $\lambda$: moderate values best complement value-based selection, while overly large $\lambda$ makes the energy dominate and biases toward plans that are easy to satisfy under the dynamics prior rather than those that maximise return. To ensure a consistent evaluation protocol, we use a single fixed hyperparameter setting in all main experiments: $K = 10$, $\mathcal{P} = 0.8$, and $\lambda = 0.1$. Unless stated otherwise, all results reported in Tables 1–2 use this configuration for all tasks and do not tune per-environment.

We also ablate the SAGE architecture by comparing it to short-horizon forward models: closed-form ridge dynamics in state space, an MLP dynamics model trained with the same optimisation protocol as our AC predictor, and a ridge

model in a random latent space. Averaged over D4RL, SAGE achieves higher discrimination (AUROC $0.98 \pm 0.00$ vs. ridge $0.88 \pm 0.01$, MLP $0.88 \pm 0.01$, and random latent $0.77 \pm 0.01$); the full protocol and domain breakdown are deferred to Appendix D.1. To quatify the computational overhead induced by SAGE gating we measure wall-clock inference latency and find that enabling SAGE adds only a small overhead ($6.8\%$) relative to DV with MCSS; see Appendix D.2 for the per-task breakdown and measurement protocol.

## 5. Related Work

Many diffusion-based decision-making approaches can be decomposed into (i) a *generator* that proposes action sequences or trajectories conditioned on the current state, and (ii) a *selector* that chooses an action by scoring candidates. Trajectory diffusion planners such as Diffuser generate state–action trajectories by iterative denoising and act via receding-horizon selection with value guidance (Janner et al., 2022; Ajay et al., 2022). In parallel, diffusion has been used as an expressive policy class for offline RL, e.g., DQL, which couples a diffusion policy with Q-guided training, and subsequent work improves computational efficiency of diffusion policies (Wang et al., 2022; Kang et al., 2023). In robotics, Diffusion Policy represents visuomotor control as action diffusion with receding-horizon execution (Chi et al., 2025). SAGE is orthogonal to these generator-side improvements as we do not need to modify the diffusion generator or its training; instead we introduce a *selector-side* feasibility score that re-ranks candidates from any generator.

A growing body of work improves robustness of diffusion-based control: Diffusion plans can fail for two intertwined reasons: sampled trajectories may (i) violate explicit constraints (e.g., collisions, bounds) or (ii) drift implicitly off the dataset's dynamical support, producing plans that look plausible but execute unreliably. Prior work can be read as intervening at different points along the generate $\rightarrow$ correct $\rightarrow$ select pipeline. The most direct intervention is hard feasibility gating where we enforce a feasibility filter via hand-coded checks when available, or via learned feasible-set identification in safe offline control—and simply reject candidates outside the feasible region (Zheng et al., 2024); this is simple and strong on safety, but can be overly conservative, brittle to modeling mismatch, and is inherently hard to scale.

A second strategy refines during sampling: RGG learns a restoration-gap predictor and injects it as denoising-time guidance, steering the diffusion process away from locally infeasible segments as they emerge (Lee et al., 2023). MCTD integrate Monte-Carlo Tree Search (MCTS) and diffusion planning to selectively expanding promising trajectories while retaining the flexibility to revisit and improve

suboptimal branches (Yoon et al., 2025). A third strategy corrects manifold drift geometrically: LoMAP performs training-free, test-time projection of intermediate samples onto a locally approximated low-rank subspace defined by nearby dataset trajectories, reducing wall-crossing and other artifact plans without retraining (Lee & Choi, 2025).

Finally, a fourth strategy couples generation to learned evaluators at selection time: candidates are filtered or ranked using additional modules such as learned viability filters (Ioannidis et al., 2025) or value/Q-based selection under batch constraints (e.g., LDCQ in a latent diffusion manifold) (Venkatraman et al., 2023); relatedly, policy-side approaches regularise diffusion parameterisations with behaviour constraints to avoid out-of-support actions (Gao et al., 2025). SAGE targets a complementary failure mode to these lines: it introduces a reward-free, self-supervised *local executability* score and applies it as a soft selector-side penalty on early prefixes, preserving the base planner's generator and long-horizon critic while suppressing candidates whose initial transitions are action-conditionally inconsistent with dataset dynamics.

Self-supervised learning has also been used to strengthen critics and value learning via auxiliary objectives. Classic lines include predictive or contrastive objectives that shape the critic's representation (Khetarpal et al., 2024; Eysenbach et al., 2022), improving data efficiency and generalisation while still training value functions with TD/reward signals (Schwarzer et al., 2020; Laskin et al., 2020). In offline RL, simple self-supervision can improve Q-function approximation and robustness (Sinha et al., 2022). JEPA-style objectives similarly learn predictive, task-agnostic representations via latent-space compatibility (LeCun, 2022; Assran et al., 2023; Li et al., 2024). SAGE builds on this predictive-compatibility view, but uses it differently: rather than improving a critic, we use self-supervised latent consistency as a separate feasibility signal to gate diffusion-plan candidates at inference time, preserving the strong base planner while suppressing locally inconsistent trajectories.

## Conclusion

We introduced Self-supervised Action Gating with Energies (SAGE), an inference-time module that improves diffusion-based planning by separating local feasibility from value. SAGE learns a feasibility signal purely from offline trajectories via predictive self-supervision and uses it to re-rank sampled plans, suppressing candidates whose early prefixes are dynamically inconsistent. This design is modular and planner-agnostic: it requires no environment interaction and no retraining of the diffusion generator or critic, and it can be integrated into existing diffusion planners. Across locomotion, manipulation, and navigation benchmarks, SAGE consistently improves strong diffusion planners and pro-

vides diagnostics showing that the learned energy localises dynamics violations, supporting feasibility-aware selection as a practical path to more reliable offline planning.

## Acknowledgements

This work is supported by Microsoft Research.

## Impact Statement

This paper presents work whose goal is to advance the field of Machine Learning. There are many potential societal consequences of our work, none which we feel must be specifically highlighted here.

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

## A. Code Availability

We provide supplementary code for reproduce the main results at:

https://github.com/edluyuan/sage

## B. Benchmarking Tasks

We benchmark on standard tasks from the D4RL suite, spanning short-horizon MuJoCo locomotion, long-horizon Franka Kitchen manipulation, and maze-based navigation (Maze2D and AntMaze). Figure 7 provides rendered snapshots of the environments.

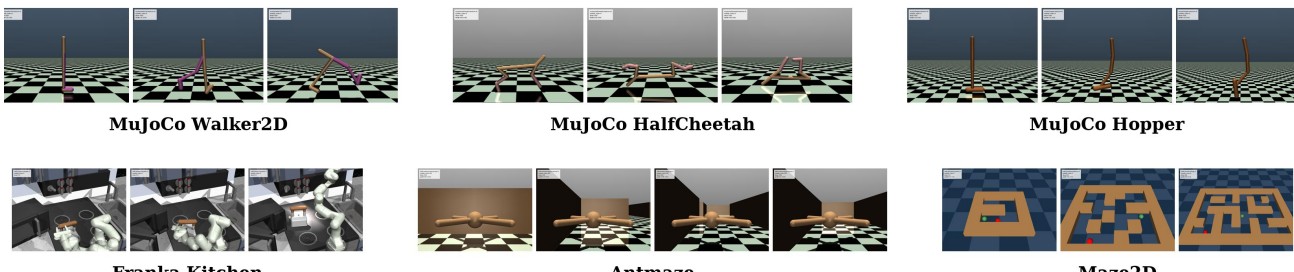

**MuJoCo Walker2D**  **MuJoCo HalfCheetah**  **MuJoCo Hopper**

**Franka Kitchen**  **Antmaze**  **Maze2D**

*Figure 7.* Rendered snapshots of the benchmark tasks used in this work. The benchmark spans MuJoCo locomotion, Franka Kitchen manipulation, and maze-based navigation (Maze2D and AntMaze).

### B.1. MuJoCo Locomotion

We evaluate continuous-control locomotion in `HalfCheetah`, `Hopper`, and `Walker2d`. These domains primarily stress local feasibility and short-horizon control: small action errors can quickly destabilise gait, reduce speed, or cause falls.

*Table 4.* **Environment details for MuJoCo locomotion experiments.**

|  | HalfCheetah | Hopper | Walker2d |
|---|---|---|---|
| State space $\mathcal{S}$ | $\mathbb{R}^{17}$ | $\mathbb{R}^{11}$ | $\mathbb{R}^{17}$ |
| Action space $\mathcal{A}$ | $\mathbb{R}^{6}$ | $\mathbb{R}^{3}$ | $\mathbb{R}^{6}$ |
| Episode length | 1000 | 1000 | 1000 |

**Datasets.** For each domain, we use the standard D4RL triad: `{medium, medium-replay, medium-expert}-v2`.

### B.2. Franka Kitchen

Franka Kitchen is a long-horizon manipulation benchmark in which a 7-DoF Franka arm interacts with multiple fixtures (e.g., cabinets, microwave, switches) within a shared scene. Unlike locomotion, success depends on sequencing and persistence over extended horizons.

*Table 5.* **Environment details for Franka Kitchen experiments.**

|  | Kitchen |
|---|---|
| State space $\mathcal{S}$ | $\mathbb{R}^{60}$ |
| Action space $\mathcal{A}$ | $\mathbb{R}^{9}$ |
| Episode length | 280 |

**Datasets.** We use `kitchen-partial-v0` and `kitchen-mixed-v0`.

## B.3. AntMaze

AntMaze couples goal-directed navigation with high-dimensional locomotion: an ant robot must traverse maze corridors to reach distant goals. This setting stresses both global planning (choosing feasible routes around walls) and local feasibility (executing turns and narrow passages without collisions).

*Table 6.* **Environment details for AntMaze experiments.**

|  | Medium-Play | Medium-Diverse | Large-Play | Large-Diverse |
|---|---|---|---|---|
| State space $\mathcal{S}$ | $\mathbb{R}^{29}$ | $\mathbb{R}^{29}$ | $\mathbb{R}^{29}$ | $\mathbb{R}^{29}$ |
| Action space $\mathcal{A}$ | $\mathbb{R}^{8}$ | $\mathbb{R}^{8}$ | $\mathbb{R}^{8}$ | $\mathbb{R}^{8}$ |
| Episode length | 1000 | 1000 | 1000 | 1000 |

**Datasets.** We evaluate four standard D4RL variants: `antmaze-medium-{play,diverse}-v2` and `antmaze-large-{play,diverse}-v2`.

## B.4. Maze2D

Maze2D is a 2D navigation benchmark where a point agent moves in a maze with walls. Because the dynamics are simple, performance is dominated by spatial reasoning and path planning under geometric constraints.

*Table 7.* **Environment details for Maze2D experiments.**

|  | UMaze | Medium | Large |
|---|---|---|---|
| State space $\mathcal{S}$ | $\mathbb{R}^{4}$ | $\mathbb{R}^{4}$ | $\mathbb{R}^{4}$ |
| Action space $\mathcal{A}$ | $\mathbb{R}^{2}$ | $\mathbb{R}^{2}$ | $\mathbb{R}^{2}$ |
| Goal space $\mathcal{G}$ | $\mathbb{R}^{2}$ | $\mathbb{R}^{2}$ | $\mathbb{R}^{2}$ |
| Episode length | 300 | 600 | 800 |

**Datasets.** We use `maze2d-umaze-v1`, `maze2d-medium-v1`, and `maze2d-large-v1`.

# C. Implementation Details

This section contains the configuration used to reproduce our results. SAGE is an inference-time, selector-side module: it does not modify the diffusion candidate generator, the critic, or any training pipeline. Accordingly, we specify three components: (i) Stage I JEPA representation learning (encoder $e_\theta$, EMA teacher $e_{\bar{\theta}}$, predictor $g_\phi$), (ii) Stage II action-conditioned latent predictor $f_\eta$, and (iii) the SAGE diffusion planner. Complete hyperparameters for each component are provided in Tables 8–11. Unless otherwise stated all training and inference was on Nvidia A100 GPUs.

## C.1. Predictive State Representation

**Data and windowing.** From offline trajectories, we sample length-$W$ state windows $s_{\text{ctx}} = (s_t, \ldots, s_{t+W-1})$. We apply two independent masking augmentations to obtain two context views $\tilde{s}_{\text{ctx}}^{(1)}$ and $\tilde{s}_{\text{ctx}}^{(2)}$ (see below), and define future target states $s_{\text{tgt}}^{(k)} = s_{t+W-1+k}$ for offsets $k \in \mathcal{K}$.

**Encodings and offset conditioning.** The encoder $e_\theta$ maps a masked context window to latent features. In our implementation, $e_\theta(\tilde{s}_{\text{ctx}}^{(i)})$ produces a sequence of per-step latents $z_{\text{ctx}}^{(i)} \in \mathbb{R}^{W \times d_z}$, which the predictor $g_\phi(\cdot, k)$ consumes together with a learned embedding of the offset $k$. The EMA teacher $e_{\bar{\theta}}$ encodes each target state into $\bar{z}_{\text{tgt}}^{(k)} = e_{\bar{\theta}}(s_{\text{tgt}}^{(k)}) \in \mathbb{R}^{d_z}$.

**Predictor readout.** The predictor $g_\phi$ is implemented as a Transformer encoder over the context latents $z_{\text{ctx}}^{(i)}$ augmented with one learnable prediction token per offset $k \in \mathcal{K}$. Each prediction token is placed at the corresponding future position via positional encoding (index $(W-1) + k$), attends to the full context, and its final hidden state is used as the aggregated representation for that offset; this vector is then combined with a learned $k$-embedding and mapped through an MLP head to produce $\hat{z}_{\text{tgt}}^{(i,k)}$.

**Alignment loss.** For view $i \in \{1, 2\}$ and offset $k \in \mathcal{K}$, the predictor outputs $\hat{z}_{\text{tgt}}^{(i,k)} = g_\phi(e_\theta(\tilde{s}_{\text{ctx}}^{(i)}), k)$. We align predicted targets to teacher targets using a stop-gradient operator $\text{sg}(\cdot)$:

$$\mathcal{L}_{\text{sim}} = \frac{1}{|\mathcal{K}|} \sum_{k \in \mathcal{K}} \sum_{i \in \{1,2\}} \left\| \hat{z}_{\text{tgt}}^{(i,k)} - \text{sg}(\bar{z}_{\text{tgt}}^{(k)}) \right\|_2^2. \tag{13}$$

**VICReg regularisation.** To prevent representational collapse without negative pairs, we add VICReg-style variance and covariance penalties on the predicted embeddings (Bardes et al., 2021). We form a batch matrix $Z \in \mathbb{R}^{\mathcal{B} \times d}$ by stacking predicted embeddings, and define per-dimension standard deviation $\sigma_j = \sqrt{\text{Var}(Z_{\cdot j}) + \epsilon}$ with $\epsilon > 0$. The variance term encourages non-trivial spread in each dimension:

$$\mathcal{L}_{\text{var}} = \frac{1}{d} \sum_{j=1}^{d} \max(0, \gamma - \sigma_j). \tag{14}$$

Let $\tilde{Z} = Z - \frac{1}{B} \mathbf{1} \mathbf{1}^\top Z$ be the centred embeddings and $C = \frac{1}{B-1} \tilde{Z}^\top \tilde{Z} \in \mathbb{R}^{d \times d}$ the covariance matrix. The covariance term discourages redundant dimensions:

$$\mathcal{L}_{\text{cov}} = \frac{1}{d} \sum_{i \neq j} C_{ij}^2. \tag{15}$$

**Overall objective.** The full Stage I loss is

$$\mathcal{L}_{\text{JEPA}} = \mathcal{L}_{\text{sim}} + \lambda_{\text{var}} \mathcal{L}_{\text{var}} + \lambda_{\text{cov}} \mathcal{L}_{\text{cov}}. \tag{16}$$

**Masking augmentations.** We apply (i) feature masking, which zeros a random subset of state dimensions per timestep, and (ii) time masking, which zeros a random subset of timesteps in the window; the two views use independent random masks. Mask ratios are reported in Table 8.

**EMA teacher update.** Teacher parameters follow an exponential moving average of the online encoder: $\bar{\theta} \leftarrow \mu\bar{\theta} + (1-\mu)\theta$, where $\mu$ follows a cosine schedule. After training the ecoder, we freeze the EMA teacher $e_{\bar{\theta}}$ and use $z = e_{\bar{\theta}}(s)$ for Stage II and inference.

*Table 8.* **Predictive State Representation Encoder architecture and optimisation.**

| Block | Hyperparameter | Value |
|---|---|---|
| *Architecture* | | |
| Encoder $e_\theta$ | MLP widths | $\dim(s) \to 512 \to 512 \to d_z$ ($d_z = 256$) |
| | Activation / norm | GELU; LayerNorm |
| Teacher $e_{\bar{\theta}}$ | Update rule | EMA of $e_\theta$ |
| | Momentum | cosine schedule $0.99 \to 0.9999$ |
| Predictor $g_\phi$ | Module | Transformer encoder |
| | Depth / heads | 2 layers; 4 heads |
| | FF dim / dropout | $4d_z$; dropout $0.0$ |
| Offset conditioning | Embedding | learned embedding for $k \in \{1, \ldots, K_{\max}\}$ |
| *Training setup* | | |
| Context window | Window / horizon | $W = 16$, $K_{\max} = 5$ |
| | Offsets per batch | $|\mathcal{K}| = 3$ |
| Augmentations | Feature mask | ratio $0.30$ |
| | Time mask | ratio $0.10$ |
| | View noise | none |
| Loss | Variance / covariance | $\lambda_{\text{var}} = 1.0$, $\lambda_{\text{cov}} = 0.1$ |
| | Other | $\gamma = 1.0$, $\epsilon = 10^{-4}$ |
| Optimisation | Optimiser | AdamW |
| | Weight decay / clip | $10^{-4}$; grad clip $1.0$ |
| | LR schedule | cosine: lr $10^{-4}$, min lr $10^{-6}$; warmup $5{,}000$ |
| | Steps / batch | $200{,}000$ steps; batch size $512$ |

## C.2. Action-Conditioned Latent Predictor

Stage II trains a short-horizon action-conditioned predictor $f_\eta$ in the frozen JEPA latent space. Given windows $(s_{t:t+W}, a_{t:t+W-1})$, we embed states into latents $z_{t:t+W} = e_{\bar\theta}(s_{t:t+W})$ and train a block-causal Transformer over time-step bundles $[z_t, a_t]$.

**Losses.**  We use the combined objective

$$\mathcal{L}_{\mathrm{AC}} = \mathcal{L}_{\mathrm{tf}} + \lambda_{\mathrm{ro}}\mathcal{L}_{\mathrm{ro}} + \lambda_{\mathrm{neg}}\mathcal{L}_{\mathrm{neg}}. \tag{17}$$

where $\mathcal{L}_{\mathrm{tf}}$ is teacher-forced one-step latent prediction, $\mathcal{L}_{\mathrm{ro}}$ is a short rollout consistency loss (horizon $H_{\mathrm{ro}}$), and $\mathcal{L}_{\mathrm{neg}}$ is a hinge term using batch-permuted actions to penalise action-insensitivity.

**Action-usage hinge.**  To penalise action-insensitive predictors, we form mismatched actions by permuting action sequences across the batch dimension. For a minibatch of size $\mathcal{B}$, sample a permutation $\pi$ of $\{1, \ldots, \mathcal{B}\}$ such that $\pi(i) \neq i$. Define the permuted action window $a'^i_{t:t+W-1} = a^{\pi(i)}_{t:t+W-1}$.

We then compute the negative predictions by running the same predictor but replacing only the action input (ground-truth latents are kept fixed):

$$\hat{z}'^i_{t+1+j} = f_\eta\big(z^i_{t+j}, a'^i_{t+j}\big), \qquad j = 0, \ldots, W-1. \tag{18}$$

The negative prediction error uses the same window length $W$:

$$E^i_{\mathrm{neg}} = \sum_{j=0}^{W-1} \big\| \hat{z}'^i_{t+1+j} - z^i_{t+1+j} \big\|_1 , \tag{19}$$

and the hinge loss is averaged across the minibatch:

$$\mathcal{L}_{\mathrm{neg}} = \frac{1}{\mathcal{B}} \sum_{i=1}^{\mathcal{B}} \big[ m - E^i_{\mathrm{neg}} \big]_+ , \qquad [x]_+ = \max(x, 0). \tag{20}$$

Intuitively, if $f_\eta$ ignores actions, then $E_{\mathrm{neg}}$ can remain small under mismatched actions, activating the hinge and encouraging action sensitivity. (see Sec. 3.2 for the main-text definitions).

Although $f_\eta$ is action-conditioned, we observed an early-stage degeneracy where the predictor can minimise teacher-forced one-step loss primarily by extrapolating from $z_t$ and treating $a_t$ as weak or ignorable input (e.g., when the frozen JEPA latents are smooth and dynamics are partly predictable from state alone). The permuted-action hinge is a lightweight regulariser that explicitly detects this failure mode: it becomes non-zero only when the prediction error under mismatched actions remains too small (i.e., $E^i_{\mathrm{neg}} < m$), indicating action-insensitivity. Importantly, this does not require drawing additional "negative samples" from a separate distribution; we reuse the same minibatch and form mismatches by in-batch permutation. In practice, the hinge is only active in the first 10 steps of training and quickly collapses to zero once $f_\eta$ begins to rely on actions, after which optimisation is dominated by $\mathcal{L}_{\mathrm{tf}}$ and $\mathcal{L}_{\mathrm{ro}}$. Thus, $\mathcal{L}_{\mathrm{neg}}$ functions as an early-stage guardrail rather than a persistent contrastive objective.

## C.3. Inference-Time SAGE Hyperparameters

At inference time, we compute the feasibility energy on the first $K$ transitions of each candidate and apply energy filtering + penalised ranking (Eqs. (11)–(12)). Unless stated otherwise, we use $K = 10$, $\mathcal{P} = 0.8$, and $\lambda = 0.1$. These hyperparameters were selected *without* using evaluation rollouts. We choose $K$ using the offline corruption and trajectory diagnostics: we select a short prefix length that reliably detects corrupted action windows in offline trajectories while avoiding overly conservative pruning of high-value candidates. We set $\mathcal{P} = 0.8$ as a conservative tail-trim to remove only the highest-energy candidates while preserving diversity. We set $\lambda = 0.1$ via scale calibration on offline candidate pools, so that $\lambda E(\hat\tau)$ acts as a modest correction to the planner score $J(\hat\tau)$ rather than dominating selection. After fixing $(K, \mathcal{P}, \lambda)$, we run the full 500-seed evaluation on all tasks with no further tuning.

*Table 9.* **Action-Conditioned Latent Predictor architecture and optimisation.**

| Block | Hyperparameter | Value |
|---|---|---|
| *Inputs and tokenisation* | | |
| Inputs | Latents / actions | $z_{t:t+W}$ ($d_z = 256$, whitened); $a_{t:t+W-1}$ |
| Tokenisation | Per-step token | bundle $[z_t, a_t]$ with type embedding + time positional embedding |
| *Model* | | |
| Backbone | Architecture | block-causal Transformer |
| | Depth / heads | 2 layers; 4 heads |
| | Hidden / dropout | hidden size 256; dropout 0.0 |
| Prediction | Target parameterisation | latent delta: $\hat{z}_{t+1} = z_t + \Delta z$ |
| Head | MLP head | $\text{LN} \to \text{Linear}(256, 512) \to \text{GELU} \to \text{Linear}(512, 256)$ |
| *Loss and optimisation* | | |
| Loss | Rollout term | $H_{\text{ro}} = 4$, $\lambda_{\text{ro}} = 1.0$ |
| | Negative term | $\lambda_{\text{neg}} = 1.0$, margin $m = 0.10$ |
| Optimisation | Optimiser | AdamW |
| | LR schedule | cosine: lr $10^{-4}$, min lr $10^{-6}$; warmup 5,000 |
| | Regularisation / clip | weight decay $10^{-4}$; grad clip 1.0 |
| | Steps / batch | 200,000 steps; batch size 256 |
| Normalisation | State / latent | state normalisation |

### C.4. Dataset Processing and Evaluation Protocol

**Normalisation.** We compute dataset statistics $(\mu_s, \sigma_s)$ and normalise states as $\tilde{s} = (s - \mu_s)/\sigma_s$ for Stage I, Stage II, and inverse dynamics. For Stage II we whiten latents under the frozen encoder.

**Training and evaluation.** Stage I and Stage II are trained purely offline and independently from the diffusion planner. We evaluate each method on 500 episode seeds per task and report D4RL normalised scores where applicable. When comparing SAGE to the DV-style baseline, we keep the same diffusion generator and scorer and modify the selection rule via SAGE.

## C.5. Diffusion Planner Design

In this section, we summarise the implementation details and design choices for the diffusion planner used in SAGE.

**Candidate generation.** At each decision step $t$, the planner samples $C$ candidate state trajectories

$$\{\hat{s}_{t:t+H}^{(i)}\}_{i=1}^{C} \sim p_\theta(s_{t:t+H} \mid s_t), \tag{21}$$

using a trajectory diffusion model and DDIM sampling. We then obtain actions via an inverse-dynamics policy $\pi_\phi(a_t \mid s_t, s_{t+1})$ (Sec. C.6). The executed action is the first action of the selected candidate under receding-horizon control.

**Generate-and-rank (MCSS).** The base planner uses Monte Carlo Samplin with Selection (MCSS): generate many candidates and rank them with a learned scorer $J(\hat{\tau})$ (implemented by the DV critic / value module). SAGE augments this ranking by adding an additional feasibility energy term, but keeps the DV scorer unchanged.

**Planner backbone and diffusion parameterisation.** We use a 1D Transformer diffusion network over time steps, predicting diffusion noise by default, trained on offline trajectories for $10^6$ gradient steps. At inference we sample with DDIM for 20 steps.

**Base-planner configuration.** Table 10 lists the full configuration used in our experiments.

*Table 10.* **Base diffusion planner configuration.**

| Setting | Value used |
|---|---|
| Guidance / selection regime | MCSS (generate $C$ candidates, rank with DV scorer $J$) |
| State–action generation | Separate (diffuse states; inverse dynamics produces actions) |
| Discount / credit assignment | $\gamma = 0.997$ |
| Planner horizon / stride | $H = 32$, stride $= 1$ |
| Candidates $C$ | 50 |
| Planner sampler / steps | DDIM, 20 denoising steps |
| Planner training steps | 1,000,000 gradient steps |
| Planner temperature | 1.0 |
| Planner predicts | Noise ($\epsilon$-prediction) |
| Planner network backbone | Transformer |
| Input embedding dim | 128 |
| Hidden size | 256 |
| Depth | 2 blocks |
| EMA rate (planner) | 0.9999; EMA used at inference |
| Batch size (planner pipeline) | 128 |

## C.6. Inverse Dynamics

In Separate generation, the diffusion model proposes a state sequence $\hat{s}_{t:t+H}$ and actions are produced by an inverse dynamics model $\pi_\phi(a_t \mid s_t, s_{t+1})$. We implement the DV-style **diffusion inverse dynamics** model: a conditional DDPM over actions, conditioned on the current and next state. At inference, we sample actions with a small number of DDPM steps and apply a temperature to control stochasticity.

*Table 11.* **Inverse dynamics / policy configuration**

| Setting | Value used |
| --- | --- |
| Inverse dynamics type | Diffusion inverse dynamics (conditional DDPM over $a_t$) |
| Conditioning | $(s_t, s_{t+1})$ (normalised states) |
| Backbone | MLP |
| Hidden size | 256 |
| Solver / diffusion steps | DDPM; 10 diffusion steps (train), 10 steps (sample) |
| Training steps | 200,000 gradient steps |
| EMA rate | 0.995; EMA used at inference |
| Sampling temperature | 0.5 |
| Learning rate | $3 \times 10^{-4}$ |

# D. Extensive Experimental Results

## D.1. Feasibility Discrimination Ablation

To test whether our feasibility energy reduces to a simple short-horizon forward model, and whether JEPA-style pretraining is beneficial even for low-dimensional states, we compare several predictors on a controlled discrimination task: distinguishing real transitions from corrupted ones.

From each offline dataset, we extract one-step transitions $(s_t, a_t, s_{t+1})$ and split episodes into train/validation sets to avoid temporal leakage. On validation batches, we form negatives by shuffling actions within the batch, $(s_t, a_{\pi(t)}, s_{t+1})$, which preserves state/action marginals while breaking action–next-state consistency.

Each method assigns an error $e(s_t, a_t, s_{t+1})$ (lower is more feasible), and we use $r = -e$ for discrimination. We evaluate: (i) **State forward (ridge)**: closed-form ridge regression predicting $\Delta s_t$ from $(s_t, a_t)$; (ii) **State forward (MLP)**: a learned MLP dynamics model trained on the same objective, normalisation, batch size, training steps. and optimisation schedule as the AC predictor, but operating directly in state space; (iii) **Random latent (ridge)**: a frozen random encoder $s \mapsto z$ with ridge regression predicting $\Delta z_t$ from $(z_t, a_t)$; (iv) **SAGE (JEPA+AC)**: our pretrained JEPA encoder and action-conditioned latent predictor, with the same latent whitening as in the main method. Errors use $\ell_1$ prediction loss in the corresponding space.

We report AUROC for classifying real (positive) vs. action-shuffled (negative) transitions, aggregated by domain as mean $\pm$ standard error across environments. Table 12 shows that SAGE yields the strongest discrimination across domains, outperforming both state-space forward models (ridge/MLP) and the random-latent baseline, indicating that JEPA representation learning and action-conditioned latent prediction are key to the feasibility signal.

*Table 12.* Feasibility discrimination (AUROC $\uparrow$) for classifying real transitions vs. action-shuffled transitions. Mean $\pm$ standard error across environments within each domain.

| Domain | State forward (ridge) | State forward (MLP) | Random latent (ridge) | SAGE (JEPA+AC) |
|---|---|---|---|---|
| MuJoCo | 0.883±0.022 | 0.927±0.018 | 0.825±0.026 | **0.980±0.003** |
| Maze2D | 0.991±0.001 | 0.983±0.001 | 0.818±0.025 | **0.993±0.001** |
| Kitchen | 0.834±0.002 | 0.646±0.003 | 0.542±0.002 | **0.987±0.000** |
| AntMaze | 0.825±0.003 | 0.805±0.004 | 0.702±0.003 | **0.940±0.001** |

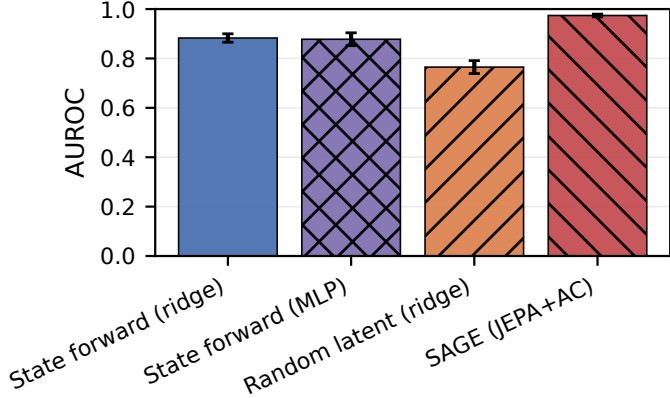

*Figure 8.* Feasibility discrimination on D4RL (AUROC $\uparrow$). Bars show mean across environments; error bars denote standard error.

## D.2. Computation Overhead at Inference

We report the wall-clock inference cost of our pipeline at test time, measured as latency per 50 vectorised environment step. We compare the MCSS baseline against our SAGE-enabled variant under identical planning and policy settings (same checkpoints, horizon $H = 10$, candidate count $C = 50$, diffusion solvers, and sampling steps).

For each vectorised step, we time the following stages: (i) **Planner:** diffusion sampling of candidate trajectories, (ii) **Critic:** scoring candidates under the baseline selector, (iii) **Action Inverse Dynamics:** action inference, e.g., diffusion inverse dynamics for executing the first action, (iv) **Environments:** environment stepping, and (v) **SAGE gating:** consisting of prefix action inference, feasibility energy evaluation, and candidate selection. Any remaining difference to the measured total is aggregated as **Other:** logging, tensor reshapes, and framework overhead.

We use a warm-up phase to avoid first-iteration CUDA/JIT effects, then measure a fixed number of consecutive steps. Specifically, We run $T_{\text{warm}} = 100$ warm-up steps followed by $T_{\text{measure}} = 500$ measured steps, recording per-step wall-clock time for each stage and for the total. We report the mean latency and the 95th percentile across measured steps. All timings are collected with identical vectorisation level (number of parallel environments) and on the same hardware/software configuration, all experiments are runned on A100 GPUs.

Appendix Figure 9 shows a stacked latency decomposition (ms per vectorised step) for four representative tasks. The hatched segment denotes the SAGE gating time. For each task, we annotate the average overhead

$$\text{overhead}(\%) \ = \ \frac{\mu_{\text{SAGE}}(T_{\text{measure}}) - \mu_{\text{MCSS}}(T_{\text{measure}})}{\mu_{\text{MCSS}}(T_{\text{measure}})} \times 100,$$

where $\mu(\cdot)$ denotes the mean across measured steps.

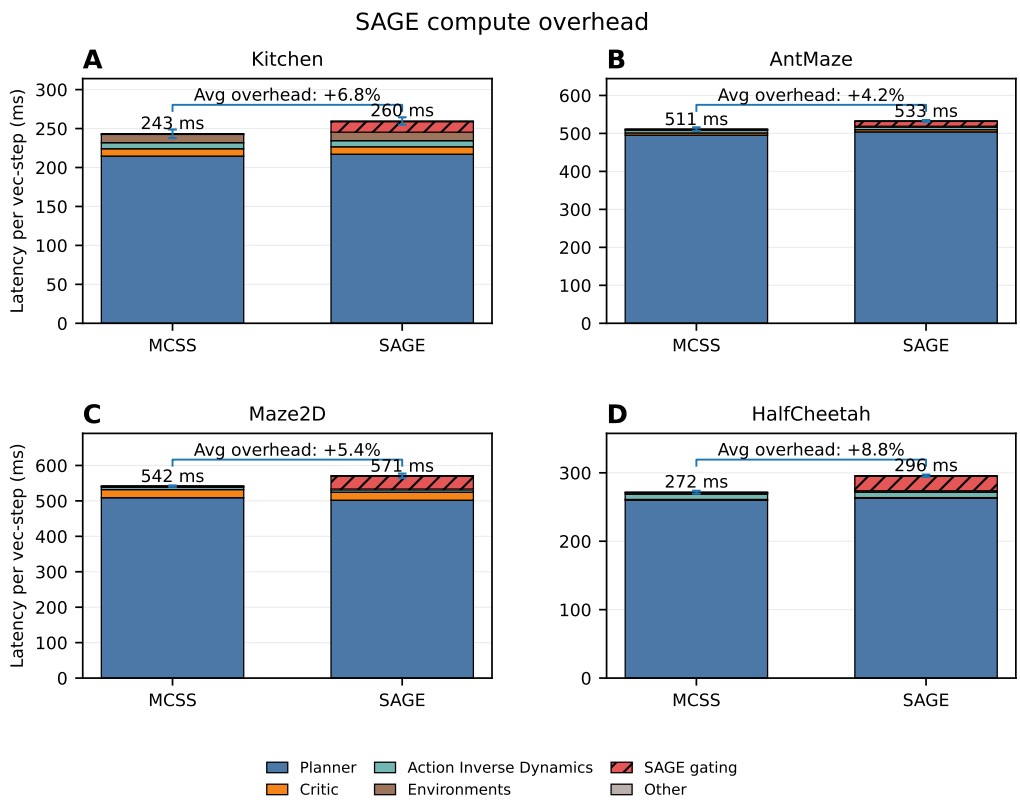

*Figure 9.* Average computational overhead of SAGE against MCSS ($C = 50$) across D4RL datasets

## D.3. Additional Feasibility Diagnostics

Figure 10 extends the controlled corruption test from the main text to a broader set of D4RL environments. Across tasks, corrupted windows consistently induce higher energies and produce localised spikes aligned with the perturbed interval, whereas energies outside the window remain close to those of the clean trajectory.

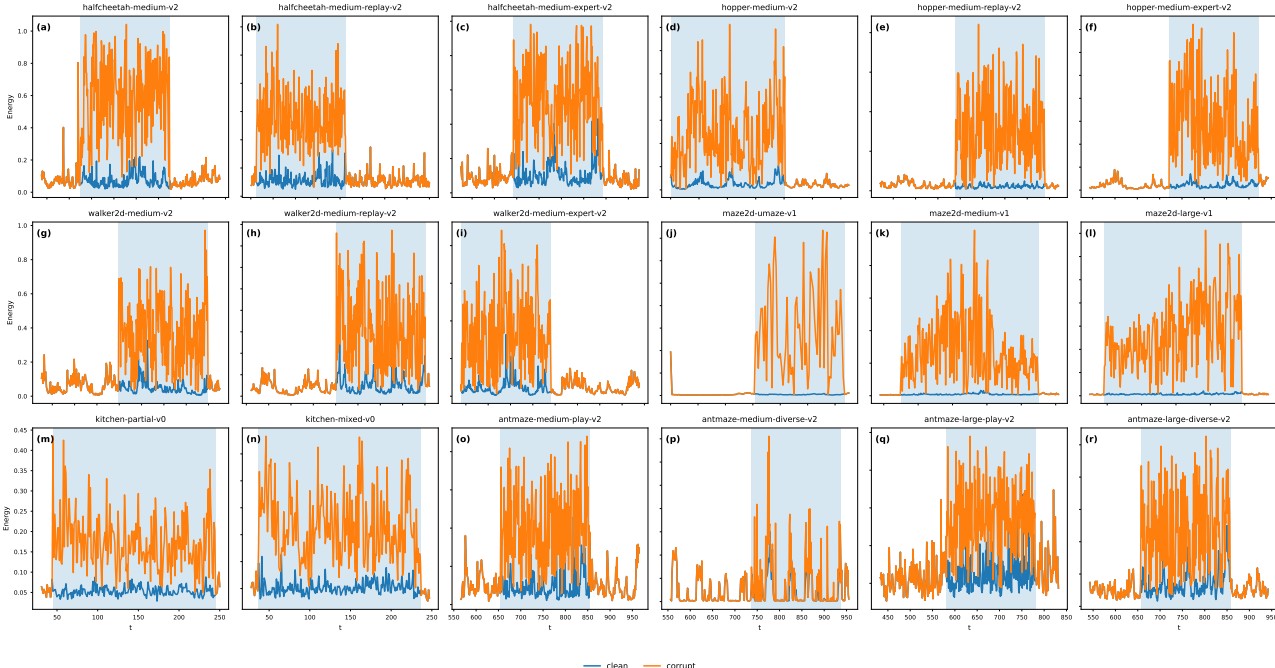

*Figure 10.* Additional feasibility energy distributions across D4RL tasks. Each panel compares clean transitions against corrupted transitions (e.g., action noise/shuffling and/or state corruption); corrupted variants consistently shift to higher energy, supporting that the SAGE feasibility signal detects locally dynamics-inconsistent samples.

### D.4. Statistical Significance

We test whether SAGE improves over the strongest diffusion-planner baseline DV using per-episode evaluation scores. For each task $i$, we evaluate each method on $N = 500$ episodes and record the D4RL normalised score $\{r_{\text{SAGE},j}^{(i)}\}_{j=1}^{N}$ and $\{r_{\text{DV},j}^{(i)}\}_{j=1}^{N}$. Unless otherwise stated, we treat episode scores as independent samples and perform an **unpaired** test of $H_0 : \mathbb{E}[r_{\text{SAGE}}] = \mathbb{E}[r_{\text{DV}}]$ against the one-sided alternative $H_1 : \mathbb{E}[r_{\text{SAGE}}] > \mathbb{E}[r_{\text{DV}}]$.

**Per-task test.** Let $\bar{r}_m$ and $s_m^2$ denote the sample mean and variance of method $m \in \{\text{SAGE}, \text{DV}\}$ on a task. We compute $\Delta = \bar{r}_{\text{SAGE}} - \bar{r}_{\text{DV}}$ and use Welch's $t$-test to allow unequal variances:

$$
t = \frac{\Delta}{\sqrt{\frac{s_{\text{SAGE}}^2}{N} + \frac{s_{\text{DV}}^2}{N}}}, \qquad \nu = \frac{\left(\frac{s_{\text{SAGE}}^2}{N} + \frac{s_{\text{DV}}^2}{N}\right)^2}{\frac{(s_{\text{SAGE}}^2/N)^2}{N-1} + \frac{(s_{\text{DV}}^2/N)^2}{N-1}}, \tag{22}
$$

where $\nu$ is the Welch–Satterthwaite degrees of freedom. We report the one-sided $p$-value under $t_\nu$ and a 95% confidence interval for $\Delta$.

**Domain and overall aggregation.** To summarise by domain $\mathcal{T}$ and overall, we aggregate across tasks using the task-average improvement

$$
\Delta_{\mathcal{T}} = \frac{1}{|\mathcal{T}|} \sum_{i \in \mathcal{T}} \left( \bar{r}_{\text{SAGE}}^{(i)} - \bar{r}_{\text{DV}}^{(i)} \right). \tag{23}
$$

We estimate uncertainty in $\Delta_{\mathcal{T}}$ using a nonparametric bootstrap over evaluation episodes: for each task $i$, we resample the $N$ episode scores with replacement independently for SAGE and DV, compute the resampled means $\bar{r}_{\text{SAGE}}^{(i)}$ and $\bar{r}_{\text{DV}}^{(i)}$, and form $\Delta_{\mathcal{T}}$ for each bootstrap replicate. We report the 95% percentile bootstrap confidence interval and a one-sided $p$-value from the bootstrap distribution. This unpaired bootstrap is conservative when evaluation episodes are matched across methods.

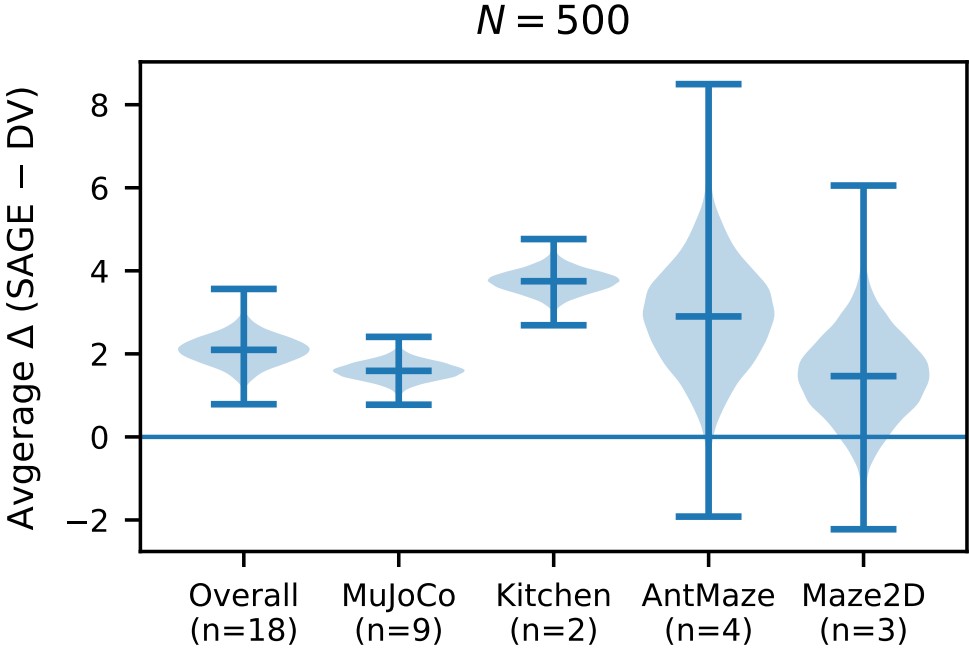

*Figure 11.* **Bootstrap distribution of SAGE−DV improvement.** Violin plots show the nonparametric bootstrap distribution (10,000 resamples) of the task-average improvement $\Delta_{\mathcal{T}}$ (SAGE−DV) computed from $N = 500$ per-episode D4RL normalised scores per task. We report results overall and by domain (MuJoCo, Kitchen, AntMaze, Maze2D). The horizontal line marks $\Delta_{\mathcal{T}} = 0$.

## D.5. Full D4RL Results

Table 13 reports the full D4RL benchmark results corresponding to the summarised tables in the main text, including Mean $\pm$ Standard Error (SE) for runs evaluated by us. All reproduced methods (marked with $^*$) are implemented and evaluated under a consistent pipeline (CleanDiffuser) with matched architectures and evaluation settings, and we use **500 evaluation episode seeds per task** throughout. Standard errors are computed across the 500 episode returns for each task. For baselines reported from the literature, we reproduce the published point estimates when available; standard errors may be absent in the original sources and are therefore not always shown here. A dash (—) indicates that a method did not report results for that environment.

*Table 13.* **Normalised performance of various offline-RL methods on the D4RL benchmark.** Reported numbers are Mean ± Standard Error over 500 evaluation seeds (when available). * denotes results reproduced by us; remaining numbers are reported from the literature, — indicates the task was not reported in the original work.

| Dataset | Environment | IL | Non-Diffusion Policies | | | Diffusion Policies | | | Diffusion Planners | | | | |
|---|---|---|---|---|---|---|---|---|---|---|---|---|---|
| | | BC | BCQ | CQL | IQL | DQL* | IDQL* | Diffuser* | RGG | LoMAP | LDCQ | DV* | SAGE (Ours) |
| Medium-Expert | HalfCheetah | 35.8 | 64.7 | 62.4 | 86.7 | 95.5 ± 0.1 | 95.9 | 88.9 ± 0.3 | 90.8 ± 0.3 | 91.1 ± 0.2 | 90.2 ± 0.9 | 92.7 ± 0.3 | 95.4 ± 0.1 |
| Medium-Expert | Hopper | 111.9 | 110.9 | 98.7 | 91.5 | 111.1 ± 1.4 | 108.6 | 103.3 ± 1.3 | 109.6 ± 2.3 | 110.6 ± 0.3 | 113.3 ± 0.2 | 108.8 ± 0.5 | 111.3 ± 0.5 |
| Medium-Expert | Walker2d | 6.4 | 57.5 | 110.0 | 109.6 | 111.6 ± 0.0 | 112.7 | 106.9 ± 0.2 | 107.8 ± 0.1 | 109.2 ± 0.1 | 109.3 ± 0.4 | 108.6 ± 0.0 | 109.4 ± 0.1 |
| Medium | HalfCheetah | 36.1 | 40.7 | 44.4 | 47.4 | 52.3 ± 0.2 | 51.0 | 42.8 ± 0.3 | 44.0 ± 0.3 | 45.4 ± 0.1 | 42.8 ± 0.7 | 50.4 ± 0.0 | 51.6 ± 0.0 |
| Medium | Hopper | 29.0 | 57.5 | 58.0 | 66.3 | 96.5 ± 1.3 | 65.5 | 74.3 ± 1.4 | 82.5 ± 4.3 | 93.7 ± 1.5 | 66.2 ± 1.7 | 80.9 ± 1.2 | 83.9 ± 1.2 |
| Medium | Walker2d | 6.6 | 53.1 | 79.2 | 78.3 | 86.8 ± 0.2 | 85.5 | 79.6 ± 0.6 | 81.7 ± 0.5 | 79.9 ± 1.2 | 69.4 ± 3.5 | 82.8 ± 0.1 | 84.8 ± 0.1 |
| Medium-Replay | HalfCheetah | 38.4 | 38.2 | 46.2 | 44.2 | 47.9 ± 0.0 | 45.9 | 37.7 ± 0.5 | 41.0 ± 0.2 | 39.1 ± 1.0 | 41.8 ± 0.4 | 45.8 ± 0.1 | 46.5 ± 0.2 |
| Medium-Replay | Hopper | 11.3 | 33.1 | 48.6 | 94.7 | 101.6 ± 0.0 | 92.1 | 93.6 ± 0.4 | 95.2 ± 0.5 | 97.6 ± 0.6 | 86.3 ± 2.5 | 91.6 ± 0.0 | 91.8 ± 0.0 |
| Medium-Replay | Walker2d | 11.8 | 15.0 | 26.7 | 73.9 | 98.2 ± 0.1 | 85.1 | 70.6 ± 1.6 | 78.3 ± 4.4 | 78.7 ± 2.2 | 68.5 ± 4.3 | 84.1 ± 0.5 | 85.3 ± 0.3 |
| **Average** | | 31.9 | 52.0 | 63.9 | 77.0 | **89.1** | 82.1 | 77.5 | 81.2 | 82.8 | 81.6 | 82.9 | 84.4 |
| Mixed | Kitchen | 47.5 | 8.1 | 51.0 | 51.0 | 55.1 ± 1.6 | 66.5 | 52.5 ± 2.5 | — | — | 62.3 ± 0.5 | 73.6 ± 0.1 | 74.5 ± 0.3 |
| Partial | Kitchen | 33.8 | 18.9 | 49.8 | 46.3 | 65.5 ± 1.4 | 66.7 | 55.7 ± 1.3 | — | — | 67.8 ± 0.8 | 90.0 ± 0.4 | 96.6 ± 0.2 |
| **Average** | | 40.7 | 13.5 | 50.4 | 48.7 | 60.3 | 66.6 | 54.1 | — | — | 65.1 | 81.8 | **85.6** |
| Antmaze-Large | Diverse | 0.0 | 2.2 | 61.2 | 47.5 | 70.6 ± 3.7 | 67.9 | 27.3 ± 2.4 | — | 39.3 ± 2.5 | 57.7 ± 1.8 | 76.0 ± 1.8 | 77.0 ± 1.7 |
| Antmaze-Large | Play | 0.0 | 6.7 | 53.7 | 39.6 | 81.3 ± 3.1 | 63.5 | 17.3 ± 1.9 | — | 20.7 ± 3.8 | — | 76.4 ± 2.0 | 82.1 ± 1.9 |
| Antmaze-Medium | Diverse | 0.0 | 0.0 | 15.8 | 70.0 | 82.6 ± 3.0 | 84.8 | 2.0 ± 1.6 | — | 36.0 ± 3.7 | 68.9 ± 0.7 | 85.1 ± 1.3 | 88.0 ± 1.7 |
| Antmaze-Medium | Play | 0.0 | 0.0 | 14.9 | 71.2 | 87.3 ± 2.7 | 84.5 | 6.7 ± 5.7 | — | 40.7 ± 4.3 | — | 89.0 ± 1.6 | 91.0 ± 2.0 |
| **Average** | | 0.0 | 2.2 | 36.4 | 57.1 | 80.5 | 75.2 | 13.3 | — | 34.2 | — | 81.6 | 84.5 |
| Maze2D | Large | 5.0 | 6.2 | 12.5 | 58.6 | 186.8 ± 1.7 | 90.1 | 123.0 | 148.3 ± 1.4 | 151.9 ± 2.7 | 150.1 ± 2.9 | 197.4 ± 1.6 | 200.6 ± 1.4 |
| Maze2D | Medium | 30.3 | 8.3 | 5.0 | 34.9 | 152.0 ± 0.8 | 89.5 | 121.5 | 130.0 ± 0.9 | 131.0 ± 0.9 | 125.3 ± 2.5 | 150.7 ± 1.1 | 150.8 ± 1.0 |
| Maze2D | Umaze | 3.8 | 12.8 | 5.7 | 47.4 | 140.6 ± 1.0 | 57.9 | 113.9 | 128.3 ± 0.8 | 126.0 ± 0.3 | 134.2 ± 4.0 | 136.8 ± 1.3 | 137.9 ± 1.0 |
| **Average** | | 13.0 | 9.1 | 7.7 | 47.0 | 159.8 | 79.2 | 119.5 | 135.5 | 136.3 | 136.5 | 161.6 | **163.1** |

