# OpenReview forum: "Improving Diffusion Planners by Self-Supervised Action Gating with Energies"
_ICML.cc/2026/Conference — ICML 2026 regular_

### Official Review · Reviewer_GRat · 2026-02-20

**Soundness:** 2
**Presentation:** 3
**Significance:** 3
**Originality:** 3
**Overall Recommendation:** 4
**Confidence:** 4

**Summary:**

The author proposes SAGE, an inference-time method for diffusion planners to enhance feasibility. The core motivation of this paper is that standard valude-guided candidate selection can choose adversarial plans that are locally inconsistent with feasible dynamics, leading to brittle execution. To address this issue, SAGE utilizes a JEPA encoder and computes a latent consistency energy to filter out infeasible plans. Experimental evaluations conducted across on D4RL benchmarks demonstrate consistent improvements over the prior diffusion-based planners.

**Compliance With Llm Reviewing Policy:**

Affirmed.

**Final Justification:**

In light of aurhors' clarifications and the newly provided results, I view the contributions of this paper more positively than in my initial evaluation. Accordingly, I will revise my score to 4.

**Key Questions For Authors:**

1. Is SAGE applicable to diffusion planners that use classifier guidance or other guided sampling strategies, rather than the unguided MCSS?
2. Beyond the three inference-time hyperparameters, the action-conditioned latent predictor introduces several training hyperparameters. How were these values determined? How much effort is required to tune them? If SAGE were applied to a new domain or task suite, would the default settings transfer, or would extensive re-tuning be necessary?

**Limitations:**

The authors do not explicitly discuss the limitations of their work in the main text. Limitations that could be acknowledged include: (1) the dependency on inverse dynamics model quality, and (2) the limited evaluations on D4RL benchmarks.

**Strengths And Weaknesses:**

Strengths:
1. The method and experimental analysis are straightforward and simple.
2. The paper is well-structured and the proposed idea is well-motivated.
3. The experimental results include comparisons with recent baselines such as LoMAP, DV, and RGG.

Weaknesses:
1. There should be many hyperparameters in SAGE which will significanly affect the performance. The paper provides ablations only for the three inference-time hyperparameters, but the action-conditioned latent predictor itself introduces additional training hyperparameters for which no ablation is provided.
2. Despite the need to tune these numerous hyperparameters, the performance gains over DV remain marginal. These margins raise questions about the practical significance of the contribution.
3. The paper claims SAGE "can integrate into existing diffusion planning ...," but all experiments are based solely on DV with MCSS. It is unclear how SAGE interacts with guided sampling methods (e.g., Diffuser and Decision Diffuser).


Minor Weaknesses:
1. The evaluation is limited to D4RL. While D4RL is widely used, it is becoming saturated in several tasks. More recent benchmarks such as OGBench would better test SAGE's scalability and practical value. However, I do not consider this a significant weakness.
2. While this paper focuses on enhancing the feasibility of generated plans, there are other active areas of research on diffusion planners, including inference-time scalability, compositionality, and long-horizon planning. I suggest the authors broaden the related work section to cover these concurrent directions [1,2,3,4].


[1] "Monte Carlo Tree Diffusion for System 2 Planning", 2025

[2] "Generative Trajectory Stitching through Diffusion Composition", 2025

[3] "State-Covering Trajectory Stitching for Diffusion Planners", 2025

[4] "Extendable Long-Horizon Planning via Hierarchical Multiscale Diffusion", 2025

---

> ### Author Rebuttal · Authors · 2026-03-30
>
> We thank reviewer`GRat` for for the clear summary and for raising useful questions about tuning burden, practical significance, and applicability beyond `MCSS`.
>
> ### Weaknesses:
> **w1** We agree that, beyond the three inference-time parameters, the action-conditioned latent predictor introduces additional training hyperparameters. In our implementation, however, these are **not tuned per task**. The training setup largely follows standard post-training recipes from prior `JEPA`-style models (e.g., learning-rate schedules), and we use a **single configuration across all domains**. In practice, we did not perform per-environment hyperparameter search, nor did we find such tuning necessary to stabilise training. We will clarify this more explicitly in the revision to better distinguish core algorithmic components from standard training choices.
>
> **w2** We also respectfully disagree with the characterization that the gains are merely marginal. We report results **over entensive 500 evaluation seeds per task**, and the improvement over `DV` is **statistically significant** and **consistent** overall (+2.10, 95% CI [1.42, 2.78]) and within `MuJoCo`, `Kitchen`, and `AntMaze`, with only `Maze2D` attenuated by ceiling effects. We will make this evidence more prominent in the final version.
>
> **w3** To further test generality, we additionally apply the SAGE selector-stage mechanism to Diffuser and Decision Diffuser(DD), again evaluating all results over 500 random seeds. In these settings, the procedure is not MCSS in the unconditional sense used for DV. Instead, for a fixed conditioning context, we sample multiple candidate trajectories, apply SAGE to penalize/filter locally infeasible state-action transitions, and then use the critic to select the candidate. The number of sampled candidates is matched to the MCSS setting in SAGE and DV for fairness, but because the sampling is conditioned rather than unconditional, we refer to this as a conditioned multi-sample selection setting rather than MCSS. The results show that, as long as the base planner is not already near collapse on a task, sampling multiple candidates under the same conditioning context and applying SAGE-based reranking generally improves performance.
> |Datasets|Diffuser*|Diffuser+SAGE|DD*|DD+SAGE|DV*|SAGE|
> |:-:|:-:|:-:|:-:|:-:|:-:|:-:|
> |Mujoco Average|$77.5$|$79.5$|$79.4$|$80.9$|$82.9$|$\mathbf{84.4}$|
> |Kitchen Average|$54.1$|$62.6$|$64.8$|$67.6$|$81.8$|$\mathbf{85.6}$|
> |AntMaze Average|$13.3$|$31.2$|$3.0$|$12.1$|$81.6$|$\mathbf{84.5}$|
> |Maze2D Average|$119.5$|$130.9$|$121.5$|$129.1$|$161.6$|$\mathbf{163.1}$|
>
> The per-task detailed result are listed in the figure here ->:  **[[greater_sage](https://hackmd.io/_uploads/Sk4BPpDiZe.jpg)]**
>
>
> **mw1** We agree that broader evaluation on newer benchmarks would strengthen the paper. We chose D4RL because it remains the most established benchmark for fair and systematic comparison with prior diffusion planners as of 2026. We view evaluation on OGBench, video datasets, and real-robot settings as important future work.
> **mw2** We appreciate this suggestion and agree that the related work section should better cover concurrent directions in diffusion planning. We will expand it to include recent works.
>
> ### Questions:
> **q1** We have added addition experiment in `SAGE` with `Diffuser` (**classifier-guided sampling**) and `Decison Diffuser` (**classifier-free guidance**) to address **w3**, results show **consistent** improment across D4RL tasks. This should vaild `SAGE` as an **effective** and **modular** improvement beyond `MCSS`
>
> **q2** We agree that, beyond the three inference-time parameters, the action-conditioned latent predictor introduces additional training hyperparameters. In our implementation, however, these are not tuned per task. The training setup largely follows standard post-training recipes from prior JEPA-style models (e.g., learning-rate schedules), and we use a single configuration across all domains.
>
> These design choices are also guided by the intended role of SAGE: it is a lightweight module for detecting short-horizon local infeasibility, rather than a large-capacity model that competes with the planner or critic. Accordingly, the network size and rollout-loss design are kept simple and fixed across tasks.
>
> In practice, we **did not perform per-environment hyperparameter search**, nor did we find such tuning necessary to stabilise training. The **same settings transfer across the D4RL domains** used in the paper. We will clarify this in the revision to better distinguish core algorithmic components from standard training choices.
>
> We again thank the reviewer for the constructive comments, which has helped us to significantly improve the manuscript. We welcome further discussions and are delighted to address all potential remaining concerns.

---

> > ### Author Rebuttal · Reviewer_GRat · 2026-04-01
> >
> > Thank you for the thoughtful rebuttal. I appreciate the authors’ detailed clarifications regarding Weaknesses 1 and 2, which have adequately addressed my concerns. I do not have further questions on these points.
> >
> > Regarding Weakness 3, the additional experimental results are convincing and effectively demonstrate the applicability of the proposed method beyond MCSS, including both unconditional and conditional diffusion planners.
> >
> > In light of these clarifications and the newly provided results, I view the contributions of this paper more positively than in my initial evaluation. Accordingly, I will revise my score to 4.

---

> > > ### Author Response · Authors · 2026-04-01
> > >
> > > We sincerely thank reviewer `GRat` for the thoughtful feedback and for engaging with our rebuttal! We greatly appreciate your careful reading and constructive feedback, which helped us improve the our work. We are glad that our clarifications and additional results addressed your concerns.

---

### Official Review · Reviewer_fHLo · 2026-03-03

**Soundness:** 3
**Presentation:** 3
**Significance:** 2
**Originality:** 2
**Overall Recommendation:** 3
**Confidence:** 4

**Summary:**

This paper addresses the execution brittleness of diffusion-based planners in offline reinforcement learning and proposes Self-supervised Action Gating with Energies (SAGE). Existing diffusion planners generate multiple candidate trajectories and select among them using a learned value function; however, this value-centric selection can favor trajectories whose early prefixes are dynamically infeasible. To address this limitation, SAGE introduces a self-supervised feasibility signal that is explicitly separated from value estimation. The method first learns a predictive latent state representation using a JEPA-based encoder trained on offline state sequences, and then trains an action-conditioned predictor to model short-horizon transitions in the learned latent space. At inference time, SAGE computes a latent prediction error as an energy score over the early prefix of each sampled candidate and combines this energy with the value score for re-ranking. The approach integrates into existing diffusion planners without retraining the generator or critic. Experiments on D4RL locomotion, navigation, and manipulation benchmarks show consistent improvements, particularly when applied to a strong generate-and-rank planner based on DV.

**Compliance With Llm Reviewing Policy:**

Affirmed.

**Final Justification:**

Thank you for the detailed rebuttal and additional clarifications. However, I will maintain my current score, as the overall performance improvements remain relatively modest, and the evaluation is limited to somewhat dated benchmarks and predominantly state-based (numeric) environments.

**Key Questions For Authors:**

## 1. Justification for JEPA-based Representation Learning

* The proposed method appears closely related to the short-horizon forward model prediction error. Could the authors clarify why a JEPA-based predictive latent representation is necessary, as opposed to using a simpler forward dynamics model trained directly in state space? The paper does not include a direct comparison against standard forward dynamics baselines. Providing empirical or theoretical justification for why JEPA-based representation learning is essential would significantly strengthen the argument for this design choice.

## 2. Comparison with Stronger Critic-Based Alternatives

* The paper argues for explicitly separating value and feasibility. However, could the observed failure mode potentially be mitigated by improving the critic itself (e.g., using ensemble Q-functions, uncertainty-aware value estimation, or stronger conservative regularization)? It would be helpful if the authors could provide either empirical comparisons or a theoretical discussion clarifying whether the proposed approach addresses a limitation that cannot be resolved by stronger value modeling alone.

## 3. Cost–Benefit Trade-off Analysis

* SAGE introduces additional training (JEPA encoder and action-conditioned predictor) and inference overhead. Could the authors provide a more systematic cost–benefit analysis, comparing the computational cost and performance gains? In particular, how does the added cost compare to training a simpler forward dynamics model or a stronger critic (e.g., ensemble Q)? A clearer comparison would help assess whether the performance improvements justify the additional complexity in practical settings.

## 4. Generalization to Visual / High-Dimensional Environments

* All experiments are conducted in state-based D4RL environments. Do the authors expect the JEPA-based latent consistency mechanism to remain stable and effective in high-dimensional visual observation settings? Providing preliminary experiments or at least a principled discussion on how the method would extend to visual environments would clarify the broader applicability of the approach.

**Limitations:**

* The paper mentions certain limitations of the proposed method, but a more systematic discussion of its scope and structural constraints would strengthen the work. In particular, the following aspects deserve clearer articulation.

* First, all experiments are conducted in state-based D4RL environments. The paper does not sufficiently discuss the potential challenges of extending the method to high-dimensional visual observation settings or real robotic environments, nor does it analyze possible failure modes in such scenarios.

* Second, although the proposed method introduces additional training and inference components, the performance improvements are relatively modest in some environments. A clearer discussion of the trade-off between computational cost and performance gains, as well as the practical implications for real-world deployment, would be beneficial.

* Third, the approach relies on approximating feasibility through short-horizon prediction error in latent space. It would be important to examine whether this approximation is sensitive to the quality, coverage, or bias of the offline dataset, and whether degraded or noisy data could adversely affect the reliability of the feasibility signal.

**Strengths And Weaknesses:**

## 1. Soundness

(1) Strengths

* The paper clearly articulates the distinction between value estimation and feasibility in the “generate-and-rank” structure of diffusion planners.

* A practical advantage is that SAGE can be integrated at inference time without retraining the diffusion generator or critic.

* The empirical evaluation is reasonably rigorous, including 500 evaluation seeds per task and statistical significance testing across multiple D4RL domains.

(2) Weaknesses

* The proposed energy function appears closely related to short-horizon forward model prediction error. It is not fully justified why JEPA-based representation learning is necessary, as opposed to simpler dynamics models.

* While the paper argues for separating feasibility from value estimation, it does not compare against stronger critic-based alternatives (e.g., critic ensembles, uncertainty-aware value functions, or stronger Q-regularization).

* There is no systematic cost–benefit analysis examining whether the performance gains justify the additional training and inference overhead introduced by SAGE.

* All experiments are conducted in state-based environments. The applicability and robustness of the method in high-dimensional visual observation settings remain unverified.

## 2. Presentation

(1) Strengths

* The problem formulation and motivation are clearly stated, and the central message that “value ≠ feasibility” is effectively communicated.

* The method is described in a structured and step-by-step manner, and the experimental setup is relatively transparent.

(2) Weaknesses

* The paper does not sufficiently justify why introducing a separate feasibility model is preferable to improving the critic itself.

* The trade-off between increased computational cost and performance improvement is not clearly analyzed in the main text.

* There is no discussion of how the proposed representation learning framework would extend to visual or high-dimensional observation settings.

* Some design choices (e.g., K = 10, specific JEPA architecture) appear largely empirical, with limited theoretical justification.

## 3. Significance

(1) Strengths

* The paper addresses a concrete and relevant weakness in diffusion-based offline RL, namely the brittleness of selection in generate-and-rank planning.

* The inference-time modularity makes the approach practically appealing.

* Consistent improvements are demonstrated in navigation and manipulation environments where feasibility issues are more pronounced.

(2) Weaknesses

* The overall performance improvements are modest, and in some environments relatively limited.

* It is unclear whether the added computational cost is justified by the magnitude of performance gains, especially for real-world applications.

* The evaluation is restricted to D4RL state-based benchmarks, with no validation in visual environments or real robotic settings.

* The method appears more as a heuristic refinement of the selection stage rather than a fundamental solution to the limitations of diffusion planners.

## 4. Originality

(1) Strengths

* The explicit separation of value and feasibility in the selector stage reflects a clear conceptual stance.

* The use of JEPA-based predictive representations for feasibility assessment is relatively novel in the context of diffusion planning.

* Focusing on improving the selection stage, rather than modifying the sampling process, differentiates this work from guidance-based approaches.

(2) Weaknesses

* At its core, the method can be interpreted as short-horizon dynamics-consistency filtering, which resembles a model-based feasibility check.

* It is unclear whether this represents a fundamentally new direction compared to improving the critic (e.g., ensemble or uncertainty-aware value estimation).

* The contribution largely consists of a combination of existing components, and may be viewed as incremental rather than algorithmically transformative.

* The work does not provide substantial new insights in representation learning or visual decision-making settings.

---

> ### Author Rebuttal · Authors · 2026-03-30
>
> We group the reviewer`fHLo`s comments into 3 recurring issues, since many of the listed weaknesses and key questions raised same underlying concerns. Specifically: meta_w1 = {Sound w2, Pre w1, Orig w2, Q2}; meta_w2 = {Sound w1, Pre w4, Orig w1, Q1, Limit 3}; meta_w3 = {Sound w3–w4, Pre w2–w3, Signif w1–w4, Q3–Q4, Limit 1–2}
>
> **meta_w1**:
> >Why explicitly separate feasibility from value, rather than only strengthening the critic?
>
> Feasibility and critic are orthogonal: critic is about higher reward; feasibility is about successful state transitions. In DV, the critic is already a strong long-horizon scorer; the failure mode we target is that value-based selection can still favor candidates whose early prefixes are dynamically inconsistent. SAGE is therefore a complementary feasibility module, not a claim that critic improvement cannot help. Note that alternatives such as ensembles or uncertainty-aware critics typically require retraining a heavier critic stack, whereas SAGE leaves the planner and critic unchanged and adds a lightweight inference-time feasibility signal. In the final version, we will make this complementarity clearer. We also added additional experiment compare against stronger critic ensemble, and forward dynamics methods as LDCP [1] and DADP [2].
> |Datasets|LDCP|DADP|SAGE|
> |:-:|-|-|-|
> |Mujoco Avg||79.7|84.4|
> |Kitchen Avg|65.2||85.6|
> |AntMaze Avg|68.4||84.5|
> |Maze2D Avg||137.8|163.1|
>
> Note that the results of LDCP and DADP are taken from the originial papers, which did not cover all the tasks. See full per-task results in the linked figure -> **[[added_baselines](https://hackmd.io/_uploads/SyqaUqwjbg.jpg)]**
>
> **meta_w2**:
> > Is SAGE a heuristic short-horizon consistency check, and are its design choices justified?
>
> This might be a misunderstanding with the characterizing SAGE as only a heuristic filter. First, our work has shown that the learned energy behaves like an action-conditioned feasibility signal: corrupting a short action window produces a sharp local energy spike and strong localisation AUROC across domains. Second, Appendix D.1 directly compares SAGE (JEPA+AC) against state-space ridge model state-space MLP, and random-latents on the same real-vs-corrupted discrimination task; SAGE is consistently strongest (AUROC 0.98±0.00 vs 0.88±0.01, 0.88±0.01, 0.77±0.01). Thus, the paper provides a direct empirical answer to Q1: the JEPA predictive latent space materially strengthens the feasibility signal beyond plain state-space forward dynamics. Likewise, $K$ is not meant to match the full task horizon. Since diffusion planning commits to the first action of the selected trajectory, SAGE only needs enough prefix to capture the early inconsistencies that matter for selection. The ablations show robustness over a moderate range of $K$; we therefore use $K=10$ across tasks to avoid per-environment tuning, not because we claim it is uniquely optimal for every domain.
>
> **meta_w3**:
> >Practical significance, cost–benefit, and evaluation scope.
>
> This is a good point, our paper has discussed this issues in the Appendix D.2, we will highlight it in the main text. Under matched conditions, enabling SAGE adds only a small avg inference overhead of 6.3%. At the same time, the main evaluation uses 500 seeds per task, and the gains over DV* are statistically significant overall (+2.10, 95% CI [1.42, 2.78]). Therefore, the fairest characterization is not “added complexity for modest gains,” but rather a lightweight inference-time add-on with modest overhead and statistically supported improvements on top of an already strong planner. On scope, D4RL remains the standard and fundamental benchmark for controlled comparison with prior diffusion planners. Extensive experiments on visual tasks or newer-benchmarks is a future step after systematically verified on D4RL fundation. As evidence beyond states, we additionally tested a V-JEPA2-AC backbone on DROID trajectories: across 20/20 sampled trajectories, action permutation increased energy (0.3463 → 0.3536, +2.11%), with 94.44% positive mean-delta steps.
>
> ### Questions:
> **q1** already addressed in Appendix D.1 via direct comparison to state-space ridge/MLP and random-latent baselines.
>
> **q2** stronger critics may help, but SAGE addresses a different, selector-side failure mode while keeping the base critic frozen.
>
> **q3** the paper already reports the latency breakdown and 500-seed significance analysis; we will surface both more prominently.
>
> **q4** please see response to reviewer`k9pg`'s **q2**. For visual-benchmark validation see meta w3.
>
> We again thank the reviewer for the constructive comments, which significantly improve the manuscript. We are delighted to have further discussions if you have any remaining concerns.
>
> [1] Venkatraman el al. Reasoning with Latent Diffusion in Offline Reinforcement Learning. ICLR 2024
> [2] Wang et al. Dynamics‑Aligned Diffusion Planning for Offline RL: A Unified Framework with Forward and Inverse Guidance. TMLR 2026

---

> > ### Author Rebuttal · Reviewer_fHLo · 2026-04-02
> >
> > Thank you for the detailed rebuttal and additional clarifications. However, I will maintain my current score, as the overall performance improvements remain relatively modest, and the evaluation is limited to somewhat dated benchmarks and predominantly state-based (numeric) environments.

---

> > > ### Author Response · Authors · 2026-04-03
> > >
> > > Dear Reviewer `fHLo`,
> > >
> > > We sincerely thank you for your time, for the discussion, and for reading our rebuttal. While we respect your final assessment, we would like to clarify two points.
> > >
> > > First, we believe the empirical improvement should be interpreted in the context of both baseline strength and evaluation rigor. Our main comparison is against an already state-of-the-art diffusion-planning baseline, and the reported gains are supported by **500-seed evaluation** and statistical testing. In this setting, achieving both **statistically significant and consistent (tested over 4 domains of 18 tasks) improvement** on top of an already highly optimized baseline like DV, with only a small additional inference cost, represents a meaningful and rigorous step forward.
> > >
> > > Second, we believe the remaining concern about "dated" and "predominantly state-based" evaluation reflects a scope mismatch more than a paper-specific weakness. This submission is a diffusion planning paper in offline RL, not a visual-control or VLA paper. In this literature, as of 2026, D4RL-style state-based benchmarks remain the standard controlled testbed for ablations and comparison with prior diffusion planners, including **most recent published works from late 2025 and 2026 [1, 2, 3, 4, 5, 6]**. For that reason, we do not believe that the absence of visual-environment experiments should be treated as a decisive weakness for the present submission, although we agree that it is an important future direction. Our preliminary DROID result was intended precisely as an initial step in that direction, rather than as the core validation setting of the paper.
> > >
> > > We again thank you for the feedback. We hope this clarification helps ensure that the paper is assessed based on its actual claims, supporting evidence, and intended problem setting.
> > >
> > > [1] Habitizing Diffusion Planning for Efficient and Effective Decision Making ICML 2025
> > > [2] Accelerating Diffusion Planners in Offline RL via Reward-Aware Consistency Trajectory Distillation ICLR 2026
> > > [3] Trajectory Generation with Conservative Value Guidance for Offline Reinforcement Learning ICLR 2026
> > > [4] Instructed Diffuser with Temporal Condition Guidance for Offline Reinforcement Learning TPAMI 2026
> > > [5] State-Covering Trajectory Stitching for Diffusion Planners NeurIPS 2025
> > > [6] Structural Information-based Hierarchical Diffusion for Offline Reinforcement Learning NeurIPS 2025

---

### Official Review · Reviewer_VzW4 · 2026-03-12

**Soundness:** 2
**Presentation:** 3
**Significance:** 2
**Originality:** 2
**Overall Recommendation:** 3
**Confidence:** 4

**Summary:**

This paper studies diffusion planners for offline reinforcement learning and highlights a key failure mode: when scoring relies purely on value functions, the planner tends to prefer trajectory prefixes that appear high-return but are inconsistent with environment dynamics, leading to brittle execution. To address this, the paper proposes SAGE. It first trains a self-supervised JEPA to learn state representations, then fits an action‑conditioned predictor in the latent space. The prediction error over the first K steps is used as a feasibility energy to filter candidate trajectories and to perform joint “value − energy” ranking. In this way, SAGE improves decision robustness without modifying the original diffusion model or critic. Experiments on D4RL tasks including MuJoCo, Franka Kitchen, AntMaze, and Maze2D show that SAGE improves performance over strong baselines such as DV, with particularly notable gains on long-horizon and sparse-reward tasks.

**Compliance With Llm Reviewing Policy:**

Affirmed.

**Final Justification:**

Thanks to the authors and reviewers. My score stands.

**Key Questions For Authors:**

- The claim that “feasibility should be modelled independently from value” seems more like a design choice than a necessary principle. Is there a reason it must be modelled separately? Given that alternative approaches use multi-head networks or multi-task learning to model feasibility and value within a single critic, and even use feasibility to regularise value learning, why is a fully separated design preferable here?
- Relative to DV, could SAGE’s gains be driven mainly by filtering out extreme “wall-crossing / physically impossible” trajectories? For tasks already near a performance ceiling, where improvements are small, and for tasks with large gains, would DV approach SAGE’s performance if one manually discarded clearly infeasible trajectories? More broadly, does the method mainly prevent extreme failures, or does it also improve performance among already feasible strategies? Can this paper disentangle these two effects more clearly?
- The choice of prefix length K appears fairly heuristic: the authors fix K = 10 based on a small number of diagnostic tasks and reuse this value across all environments. While this avoids per-task tuning, is it appropriate for tasks with very different time scales (e.g., Hopper vs. Kitchen vs. AntMaze), where the optimal local window length might naturally differ? Could performance be sensitive to K, and if so, how robust is the method to this hyperparameter?

**Limitations:**

see weakness

**Strengths And Weaknesses:**

Strengths
- The Related Works section reorganises diffusion RL methods along the generate, correct and select pipeline, which clearly situates the proposed method within the existing ecosystem. The paper is structurally complete, making it relatively easy to reproduce and follow.
- The method is cleanly modularised and compatible with strong existing baselines. Concretely, it augments existing MCSS-style diffusion planners such as DV by adding a feasibility filtering/penalty layer only in the selection stage, and can be plugged into any diffusion planner that can generate candidate trajectories and provide a scorer, without changing the planner architecture itself.
- Experiments cover multiple D4RL domains (MuJoCo, Kitchen, AntMaze, Maze2D), report domain-wise average gains and significance statistics, and compare against a broad set of baselines spanning traditional offline RL, diffusion policies, and various diffusion-planner families.
Weaknesses and limitations
- Using only the first K steps’ latent prediction error to assess feasibility misses violations that only emerge in the mid or late parts of the horizon. When failures arise after step H to K, SAGE may not detect them.
- The quality of the energy signal depends heavily on the JEPA state representation. A relatively heavy JEPA is used even in low-dimensional domains, but there is no systematic analysis of failure modes when representation learning is poor.
- Empirical evaluation relies primarily on D4RL scores and lacks systematic metrics of “catastrophic failures” (e.g., collision counts, out-of-bounds rate, fraction of infeasible actions). Evidence that SAGE reduces clearly impossible behaviours such as wall-crossing is mostly qualitative or illustrative.

---

> ### Author Rebuttal · Authors · 2026-03-30
>
> **w1** We would like to clarify that this aspect of SAGE is tied to its intended role in the planning pipeline. SAGE evaluates feasibility over the first $K$ steps because it is used **within the diffusion planner’s decision horizon**, where candidate trajectories are actually ranked and selected. In this setting, the practically relevant failure modes are early-prefix local inconsistencies that affect which candidate is executed. Our claim is therefore not to provide a full-horizon verifier, but to supply the selector with an executability signal over the part of the trajectory that matters most for action selection.
>
> **w2** The energy quality depends on the learned representation, as with any representation-based feasibility score. However, the JEPA module is a deliberate component choice rather than unnecessary overhead: Appendix D.1 shows substantially stronger feasibility discrimination than simpler state-space forward baselines, while Appendix D.2 shows only modest added latency relative to the MCSS pipeline.
>
>
> **w3** We agree that the main paper primarily reported standard D4RL scores and illustrated wall-crossing qualitatively. In response, we added an explicit collision analysis on **Maze2D-Large** and **AntMaze-Large-Diverse**. To the best of our knowledge, prior work in this line of diffusion-based planning does not provide a systematic quantitative analysis of such failure modes [1, 2, 3]. We define a collision as a planned transition whose motion segment intersects a wall cell in the maze occupancy grid; for AntMaze, the same map-based criterion is applied to the ant’s \((x,y)\) position. Across **20 sampled trajectories**, SAGE substantially reduces collisions relative to DV under both \(C=50\) and \(C=500\), providing direct quantitative evidence that SAGE suppresses catastrophic infeasible behavior beyond what is captured by return alone.
>
>
> |Datasets|Env|DV ($C=50$)|SAGE ($C=50$)|DV ($C=500$)|SAGE ($C=500$)|
> |:-:|:-:|:-:|:-:|:-:|:-:|
> |Maze2D|Large|57|5|105|9|
> |AntMaze-Large|Diverse|44|3|130|17|
>
> ### Questions:
>
> **q1** We do not claim that feasibility must always be modeled separately as a universal principle. Our claim is is that in **MCSS-style offline diffusion planning**, explicit separation is a well-motivated design choice[4]. The selector is effectively solving a **constrained ranking** problem: choose the highest-value candidate whose executable prefix is locally supported by the data. This is consistent with constrained/safe RL and with offline RL, where value critics alone are known to be fragile under support mismatch and are often augmented with support, conservatism, or uncertainty mechanisms. Since MCSS commits to the first action of the selected trajectory, early-prefix errors matter the most. Our point is therefore not that joint critics are impossible, but that a selector-side feasibility signal is a principled complement to a strong long-horizon critic.
>
> **q2** We believe that SAGE’s gains are not driven only by removing obvious wall-crossing / physically impossible trajectories. Those are merely the strongest manifestations of infeasibility. More generally, SAGE has two effects: (i) it suppresses extreme failures with very high feasibility energy, and (ii) it performs soft reranking, preferring lower-energy candidates among similarly valued trajectories. Thus, manually discarding only blatantly infeasible trajectories would address only the first effect, not the second.
>
> **q3** We agree that $K$ should be justified carefully. Our rationale is not that $K 10$ is universally optimal, but that SAGE is designed to score **local executability within the planner horizon**, and in diffusion planning the executed action is always the **first action** of the selected trajectory. Thus, the goal of $K$ is not to cover the full task timescale, but to capture enough of the early prefix to detect inconsistencies relevant for selection. In practice, we find that **moderate prefix lengths** works well across environments without extensive tuning. We intentionally report a **single shared value** across all tasks to avoid per-environment tuning and keep the comparison fair. The ablations suggest that performance is relatively stable over a moderate range of $K$, with degradation mainly at extreme choices such as $K=1$ or overly large $K$.
>
> We again thank the reviewer for the constructive comments, which has helped us to significantly improve the manuscript. We welcome further discussions and are delighted to address all potential remaining concerns.
> ### Ref：
> [1] Feng et al. Resisting Stochastic Risks in Diffusion Planners with
> the Trajectory Aggregation Tree. ICML 2024
> [2] Lee and Choi Local Manifold Approximation and Projection for Manifold-Aware Diffusion Planning. ICML 2025
> [3] Ki et al Prior-Guided Diffusion Planning for Offline Reinforcement Learning. NeurIPS 2025
> [4] Lu et al. What Makes a Good Diffusion Planner for Decision Making? ICLR 2025

---

> > ### Author Rebuttal · Reviewer_VzW4 · 2026-04-07
> >
> > Thank you for the rebuttal. The added collision analysis is helpful and strengthens the empirical case. However, my main concerns remain only partially resolved: the separation of feasibility and value is better motivated but not clearly shown to be preferable, the fixed prefix length K still appears heuristic, and the dependence on representation quality remains insufficiently analyzed. My overall assessment therefore remains unchanged.

---

> > > ### Author Response · Authors · 2026-04-08
> > >
> > > We sincerely thank the reviewer for their continued engagement and for acknowledging that our new collision analysis strengthens the empirical case for SAGE.
> > >
> > > We agree that a joint critic is a valid architectural choice in many RL paradigms. However, our claim is specifically that in MCSS-style and multi-sample diffusion planning, separating these signals offers a distinct structural advantage: modularity.
> > >
> > > To further support the generality and credibility of SAGE, we additionally apply the SAGE selector-stage mechanism to Diffuser and Decision Diffuser (DD), again evaluating all results over 500 random seeds. In these settings, the procedure is not MCSS in the unconditional sense used for DV. Instead, for a fixed conditioning context, we sample multiple candidate trajectories, apply SAGE to penalize or filter locally infeasible state-action transitions, and then use the critic to select the candidate. The number of sampled candidates is matched to the MCSS setting in SAGE and DV for fairness, but because the sampling is conditioned rather than unconditional, we refer to this as a conditioned multi-sample selection setting rather than MCSS. The results show that, provided the base planner is not already near collapse on a task, sampling multiple candidates under the same conditioning context and applying SAGE-based reranking generally improves performance.
> > >
> > > |Datasets|Diffuser*|Diffuser+SAGE|DD*|DD+SAGE|DV*|SAGE|
> > > |:-:|:-:|:-:|:-:|:-:|:-:|:-:|
> > > |Mujoco Average|$77.5$|$79.5$|$79.4$|$80.9$|$82.9$|$\mathbf{84.4}$|
> > > |Kitchen Average|$54.1$|$62.6$|$64.8$|$67.6$|$81.8$|$\mathbf{85.6}$|
> > > |AntMaze Average|$13.3$|$31.2$|$3.0$|$12.1$|$81.6$|$\mathbf{84.5}$|
> > > |Maze2D Average|$119.5$|$130.9$|$121.5$|$129.1$|$161.6$|$\mathbf{163.1}$|
> > >
> > > The detailed per-task results are provided in the linked figure below:
> > > **[[greater_sage](https://hackmd.io/_uploads/Sk4BPpDiZe.jpg)]**
> > >
> > > Second, regarding the prefix length $K$, the paper does not present $K = 10$ as universally optimal. The main text already reports cross-domain ablations over $K$, showing that performance is stable over a moderate range and degrades mainly at extreme values, and it states that a single shared $K$ is used across all tasks specifically to avoid per-environment tuning and keep the comparison fair. The rebuttal reiterated exactly this rationale.
> > >
> > > Third, regarding representation quality, we agree that any representation-based feasibility score depends on the quality of the representation. This is precisely why we do not use simple state-space models,In Appendix D.1, we already provides the direct analysis that is operationally relevant here: SAGE is compared against state-space ridge, state-space MLP, and random-latent baselines on the same feasibility-discrimination task, and the JEPA-based representation is consistently strongest across domains. This directly addresses why the paper uses a stronger predictive representation rather than a simpler or weaker one.
> > >
> > > We are grateful that the reviewer recognizes the breadth of our evaluation, the strength of the baselines, and the concrete reductions in catastrophic failures. We hope these clarifications, alongside the new evidence of SAGE's broad architectural compatibility, resolve the remaining concerns.

---

### Official Review · Reviewer_k9pg · 2026-03-16

**Soundness:** 3
**Presentation:** 3
**Significance:** 2
**Originality:** 3
**Overall Recommendation:** 4
**Confidence:** 4

**Summary:**

This paper proposes an add-on enhancements to diffusion-based offline RL. The authors presents the Self-supervised Action Gating with Energies (SAGE) framework, which reranks the state-action trajectories generated by diffusion planner models to select one that is not only valued high but also dynamically feasible. The main technical contribution is the combination of JEPA training on state sequences and a forward dynamics model in latent space. The forward prediction model includes a novel action-usage hinge objective that encourages the predictions from different actions to be different. Given the state-action trajectories, SAGE computes the consistency energy as the latent prediction error over the first K steps. Then, only the top P fraction most feasible trajectories are retained and the winner is selected as a weighted sum of the planner score and the feasibility.

Experiment evaluations consists of 2 parts. First the authors verify the efficacy of the feasibility prediction model. When actions inside a windows in a trajectory are randomly permuted, the model predicts high energy, indicating successful identification of of infeasible actions. In Maze2D, the model filters out trajectories that cut into the maze walls. Next, the authors evaluate offline RL performance in tasks from the D4RL suite. In most tasks and on average, SAGE outperforms diffusion policies, diffusion planners and other non-diffusion-based methods, including the strongest baseline "Diffusion Veteran", which SAGE builds on.

**Compliance With Llm Reviewing Policy:**

Affirmed.

**Final Justification:**

The author rebuttal clarified the technical contributions, writing ambiguities and answered my questions. Extending beyond the D4RL benchmark wouldn't be feasible in short time. Overall, my evaluation remains unchanged.

**Key Questions For Authors:**

- Figure 3 caption says 100 trajectories while the text says 500. Did I miss any details?
- How well would the feasibility predictor work if we replace the JEPA embeddings with pretrained embeddings from image foundation models such as DINO?

**Limitations:**

The authors could discuss limitations such as assumption on demonstration qualities, limited evaluation on simulation domain and the horizon of the tasks.

**Strengths And Weaknesses:**

Strengths:
- The problem setting, motivation and related literatures and explained thoroughly. This makes the position of this work clear.
- The main SAGE framework, including the JEPA representation learning and action-conditioned latent predictor are rigorously formulated and explained with pseudo code and training objective equations.
- Benchmark performance gains on top of SOTA diffusion planners are apparent across the domains in D4RL.

Weaknesses:
- The JEPA framework and dynamics learning in latent space are established approaches. The novelty being the application for trajectory ranking is limited
- D4RL is a bit old now. Beating this benchmark is not super convincing. I wonder how this approach performs in in long-horizon robot manipulation tasks such as LIBERO and in real hardware settings.

---

> ### Author Rebuttal · Authors · 2026-03-30
>
> We thank reviewer`k9pg` for the positive assessment and for the helpful questions on novelty, scope, and presentation!
>
> ### Weaknesses:
>
> **w1** We agree that the individual building blocks: JEPA pretraining and latent dynamics prediction are not themselves new. Our intended contribution is the new use of lightweight  self-supervised latent prediction as a modular, selector-side feasibility signal for diffusion planning. SAGE is designed to preserve a strong existing generator and value scorer, while adding an orthogonal notion of local executability during candidate ranking. We will revise the framing to make this contribution more precise and avoid implying novelty at the level of the individual components.
>
> **w2** We chose the 18-task, 4-domain D4RL benchmark to enable the most direct and controlled comparison with prior diffusion-planning baselines, since this is still the setting in which most baselines and recent approaches such as Diffuser, DV, LoMAP, and other related approaches are most consistently reported. We agree that evaluation on newer benchmarks such as `LIBERO`, `OGBench`, and ultimately real-hardware settings would further strengthen the paper. However, diffusion-planning baselines are currently much less standardized in those settings, espeically for the most recent works, and real-robot evaluation also introduces substantial system and resource constraints that would make comparison less controlled. We will therefore clarify that our empirical scope is intentionally centered on D4RL for fair benchmarking, and present broader evaluation on long-horizon robot manipulation and hardware as important future work.
>
>
>
> ### Questions:
> **q1** Regarding Figure 3, thank you for catching the ambiguity! In the `MCSS` stress test, each `MCSS` run samples $C = 500$ candidate trajectories; `SAGE` then penalizes/filters infeasible ones, and the critic selects one trajectory for execution. The figure visualizes $100$ selected trajectories from $100$ repeated `MCSS` runs (i.e., one per run), not all $500$ MCSS candidates from a single run. We will revise the caption/text to make this distinction explicit.
>
> **q2** We agree that replacing JEPA embeddings with other pretrained visual representations (e.g., DINO-style features) is a promising direction, since SAGE only requires a representation in which action-conditioned local consistency can be evaluated. As preliminary evidence, we tested a vision-based JEPA backbone (finetuned V-JEPA2-AC) on DROID by comparing each trajectory’s energy under the true action sequence versus 99 randomly permuted action sequences: the effect was positive for all 20/20 trajectories, with energy increasing from 0.3463 to 0.3536 (+2.11% mean paired increase), and 94.44% of steps showing a positive mean delta. This effect was statistically significant under both a one-sided sign test and Wilcoxon signed-rank test ($p=9.54\times10^{-7}$), supporting that the method can extend beyond state-based representations.
>
> | Original | Permuted | Mean paired increase | Steps with positive mean delta |
> |:--------:|:--------:|:--------------------:|:------------------------------:|
> |  0.3463  |  0.3536  |        +2.11%        |             94.44%             |
>
> To illustrate this qualitatively, we also include one randomly selected trajectory showing the energy of the true trajectory, the mean energy under 99 permuted action sequences, and the accumulated energy delta in the figure here -> **[[VJEPA_DROID](https://hackmd.io/_uploads/BkmRVVzsbx.png)]**
> We again thank the reviewer for the constructive comments, which has helped us to significantly improve the manuscript. We welcome further discussions and are delighted to address all potential remaining concerns.

---

> > ### Author Rebuttal · Reviewer_k9pg · 2026-04-04
> >
> > Thanks to the authors for the detailed rebuttal!
> >
> > Your justification for sticking to the D4RL benchmark to maintain standardized comparisons with existing diffusion-planning baselines is reasonable. I agree that running full and fair evaluations on newer benchmarks like LIBERO or real hardware wouldn't be feasible during a short rebuttal period.
> >
> > Thanks also for addressing my other points, specifically: clarifying the intended novelty, clearing up the confusion in Figure 3, and the extra preliminary experiments.
> >
> > You have adequately addressed my questions and concerns, so I will be maintaining my score of Weak Accept.

---

> > > ### Author Response · Authors · 2026-04-04
> > >
> > > Thank you very much for the careful follow-up and for noting that our rebuttal adequately addressed your questions and concerns. We really appreciate your time and thoughtful engagement!
> > > If helpful, we just wanted to note that the acknowledgement label may be slightly inconsistent with your written comment, since your message indicates the concerns were addressed and that you maintain the current score. But in any case, we are very grateful for your support and feedback!

---

### Decision · Program_Chairs · 2026-04-30

**Decision:**

Accept (regular)

**Comment:**

This paper proposes SAGE, an inference-time add-on for diffusion planners that re-ranks sampled trajectories using a self-supervised feasibility energy (JEPA + action-conditioned latent predictor) to penalize short-horizon dynamical inconsistency, and combines this with value-based selection.

Reviewers agree the method is clean, modular, and the empirical study is careful, showing consistent, statistically significant improvements over strong diffusion-planning baselines on D4RL, with additional evidence that SAGE reduces catastrophic infeasible behaviors (e.g., wall/collision violations)

Concerns focus on incremental novelty, scope limited to D4RL/state-based tasks, and some heuristic choices (e.g., prefix length K, dependence on representation quality).

This is a borderline paper, due to the fact that the main contribution is based on combining existing techniques ( (JEPA + action-conditioned latent predictor),the limitation to (state-based) D4RL benchmark and the incremental improvement to DV in term of absolute performance (that can be due on the benchmark itself is saturated).